# ENHANCING COGNITION AND EXPLAINABILITY OF MULTIMODAL FOUNDATION MODELS WITH SELF-SYNTHESIZED DATA

**Yucheng Shi**[1]  **Quanzheng Li**[2]  **Jin Sun**[1]  **Xiang Li**[2]  **Ninghao Liu**[1]
[1]School of Computing, University of Georgia
[2]Department of Radiology, Massachusetts General Hospital and Harvard Medical School

## ABSTRACT

Large Multimodal Models (LMMs), or Vision-Language Models (VLMs), have shown impressive capabilities in a wide range of visual tasks. However, they often struggle with fine-grained visual reasoning, failing to identify domain-specific objectives and provide justifiable explanations for their predictions. To address the above challenge, we propose a novel visual rejection sampling framework to improve the *cognition* and *explainability* of LMMs using self-synthesized data. Specifically, visual fine-tuning requires images, queries, and target answers. Our approach begins by synthesizing interpretable answers that include human-verifiable visual features. These features are based on expert-defined concepts, and carefully selected based on their alignment with the image content. After each round of fine-tuning, we apply a reward model-free filtering mechanism to select the highest-quality interpretable answers for the next round of tuning. This iterative process of synthetic data generation and fine-tuning progressively improves the model's ability to generate accurate and reasonable explanations. Experimental results demonstrate the effectiveness of our method in improving both the accuracy and explainability of specialized visual classification tasks.

## 1 INTRODUCTION

Large Multimodal Models (LMMs) have recently demonstrated remarkable performance in visual descriptive tasks, excelling in applications such as image captioning and visual question answering (Liu et al., 2024a; Alayrac et al., 2022; Li et al., 2022; Liu et al., 2024b). However, recent research (Kim & Ji, 2024; Geigle et al., 2024) reveals a significant drop in their effectiveness when applied to knowledge-intensive, domain-specific visual classification tasks. We observe that even advanced models like LLaVA-1.5 (Liu et al., 2024a) can only achieve a mere 12.2% classification accuracy in the Stanford Dogs dataset (Khosla et al., 2011). Further investigation reveals that current LMMs struggle to utilize key visual features for reasoning in classification and fail to provide justifiable explanations for their predictions, as illustrated by the examples in Figure 1.

**User:** Identify the bird species and explain your reasoning.

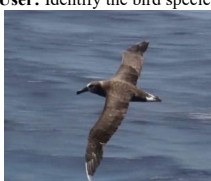

**LMM:** The bird species shown in this image is a *seagull*. The reason for this identification is that **the bird is flying over the ocean**, which is a typical habitat for seagulls.
**Label:** *Black footed Albatross*

**LMM:** The bird species is a *black crow*. The reason for this identification is that **the bird is perched on a tree branch**, which is a common behavior for crows.
**Label:** *Groove billed Ani*

Figure 1: LLaVA-1.5 struggles to utilize key visual features in images for reasoning and explaining predictions in classification tasks.

The core issue stems from insufficient domain-specific alignment, as the model struggles to recognize key visual features and link them to the correct labels. To address this problem, we propose enhancing the LMM's domain-specific cognition and explainability through *fine-tuning* (Touvron et al., 2023; Gu et al., 2021). However, this approach is hindered by a lack of data, as creating high-quality, feature-level image annotations is both complex and resource-intensive (Liu et al., 2024c). While labeling images by category and identifying key features for each class independent of the image is manageable, annotating the *specific visual characteristics per image* requires an extensive workload. Moreover, this level of detailed annotation goes beyond the capacity of standard annotators and current LMMs (Chen et al., 2024a), making it impractical to scale.

The biggest challenge now is synthesizing high-quality training data, specifically interpretable target answers. Given a dataset with images and labels, a naive approach would be to use only labels as target answers. However, training on such data may result in shortcut learning, where models pick up spurious correlations instead of truly understanding key visual features (Geirhos et al., 2020). While including general label-associated features as target answers might seem beneficial, it often results in overly generic explanations that lack the *image-specific details* necessary for accurate interpretation. We illustrate these shortcomings with examples in Figure 2.

To tackle this challenge, we propose a framework that allows LMMs to self-synthesize interpretable answers without relying on explicit image-specific annotations. For a given image, we first leverage the LMM's captioning ability to generate descriptions, which are then used to identify visual features relevant to that specific image. Each description may only cover part of the key features, but by collecting a large set of descriptions from the LMM, we can approximate the true distribution of the image's features, reducing the incompleteness in individual descriptions. We provide a formal justification for this approach in Section 3.2. Moreover, to ensure precise identification, we apply an information bottleneck technique to select the most relevant features. Once the image-specific concepts are identified, they are rewritten into interpretable answers.

For the training procedure, we also design an iterative fine-tuning approach to further improve performance over a one-shot training scheme. We begin by extracting image-level features and transforming them into interpretable answers, which, together with the corresponding images and queries, form the initial training dataset. Fine-tuning on this data results in an updated model that can generate more accurate answers. The updated model is then used to repeatedly generate answers, with the best one selected for the next round of fine-tuning. This self-boosting process progressively improves the LMM's ability to deliver reliable explanations.

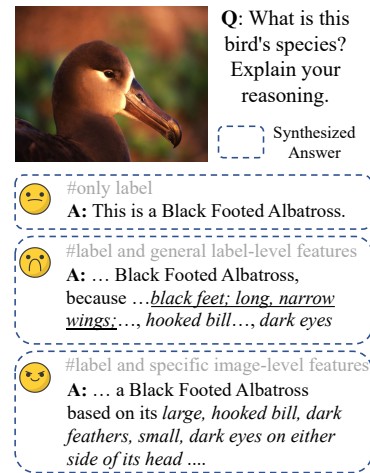

Figure 2: Examples of synthetic answers for query **Q**. Training with the first two types leads to shortcut learning or overgeneralization.

In summary, our contributions are threefold: (1) We propose a novel framework that improves LMMs' interpretable visual classification abilities without requiring extensive manual labeling, (2) We introduce an information-theoretic approach to select interpretable visual concepts for each image and a reward model-free filtering method to ensure high-quality data selection from synthetic outputs, and (3) We develop an iterative process of data synthesis and model fine-tuning to progressively enhance LMMs' cognitive abilities and explainability.

## 2 PRELIMINARY

**Problem Statement.** We aim at developing a Large Multimodal Model (LMM) for explainable visual classification. Let $f_\theta$ denote the LMM model, $X$ be the input image, and $q$ be the query prompt. The model's answer is denoted as $\hat{y} = f_\theta(X, q)$, where $\hat{y}$ is expected to correctly predict the label and explain its prediction by using the visual features observed in the image. To build such a model, a straightforward strategy is to *fine-tune* the LMM with a training dataset that contains the ground-truth answer for each input image. However, most available datasets $\mathcal{D} = \{(X_i, c_i)\}_{i=1}^N$ only consist of images $X_i$ and class labels $c_i$ (Khosla et al., 2011). To solve this, we propose a data synthesis approach that transforms the raw dataset into an augmented dataset $\mathcal{D}^* = \{(X_i, q_i, y_i)\}_{i=1}^N$ with queries and explainable answers. While generating queries $q_i$ is straightforward, the challenge lies in synthesizing explainable answers $y_i$, which must include detailed visual features $Z_i^*$ that humans can identify and use for explanations. Therefore, the key problem is developing a method to automatically annotate visual features $Z_i^*$ for each image $X_i$, given their labels $c_i$.

**Visual Fine-Tuning.** Visual fine-tuning adapts a pre-trained LMM to understand specific visual information by training on image-text pairs. Typically, an LMM $f_\theta$ consists of a vision encoder that extracts visual embeddings from an input image $X$, a projector that maps these embeddings into the language embedding space, and a language model that processes the combined visual and

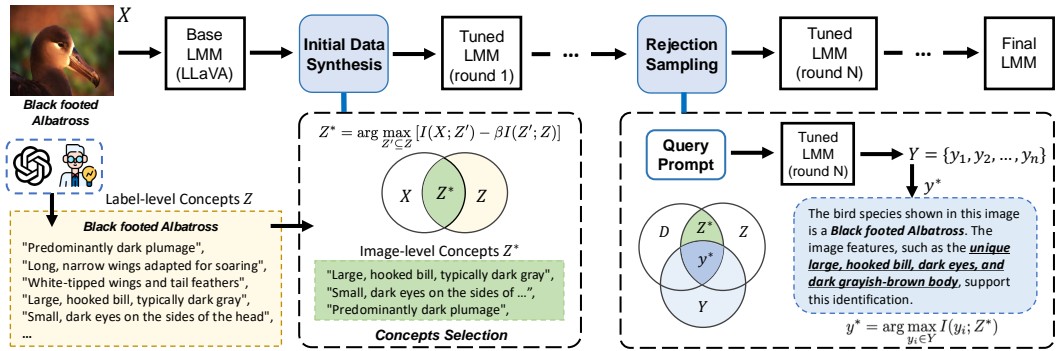

Figure 3: Our framework: An iterative approach of data synthesis and model fine-tuning.

textual information (Liu et al., 2024b;a; Lin et al., 2024). Formally, given a round of conversation containing an image $X$, a question $q$ and an answer $y$, the model is trained to maximize the likelihood of generating the target answer: $\mathcal{L}(\theta) = \sum_{i=1}^{|y|} \log p_\theta(y^i|X, q, y^{<i})$. Here, $y^i$ is the $i$-th token of the answer $y$, and $|y|$ is the length of the answer. The fine-tuning process typically optimizes performance by freezing the pre-trained visual encoder to preserve learned visual representations while updating the projector and language model parameters to improve language understanding for visual inputs (Liu et al., 2024b; Lin et al., 2024). Recognizing the critical role of fine-tuning data quality in model performance, our research proposes synthesizing high-quality conversation data to improve performance.

## 3 METHODOLOGY

### 3.1 OVERVIEW

Our approach leverages synthetic data for visual fine-tuning to enhance both the cognitive ability and explainability of LMMs, as illustrated in Figure 3. There are two major steps:

**1. Image-level Visual Concept Selection**: Given an image and its label, our first task is to extract a set of *image-specific concepts* that explain the connection between the image and label. We propose a selection method that identifies a subset of concepts most relevant to the image content while ensuring the subset is concise. Using these concepts, we prompt the LMM to rewrite them into textual answers that explain the label. These answers are used for the first round of fine-tuning.

**2. Reward Model-Free Rejection Sampling**: After the initial fine-tuning, the model can generate synthetic answers, which can be used for next-round fine-tuning, but their quality still varies. To filter out low-quality explanations and only select the best quality one as training data, we use previously selected concepts as filtering criteria, quantifying alignment between explanations and the concepts. The synthetic answer with the best-aligned explanation is then selected and combined with the corresponding image and query, forming a new *data pair* for subsequent rounds of tuning.

### 3.2 STEP 1: IMAGE-LEVEL VISUAL CONCEPT SELECTION

In this step, our goal is to identify the visual concepts present in a given image. Let $X$ represent the true content of the image and $c$ its class label. Each label class $c$ is associated with a set of expert-defined visual concepts $Z$, which can be obtained by consulting domain experts or using a large language model. However, not all concepts in $Z$ will necessarily be present in the image $X$. Therefore, we aim to select a subset of concepts, $Z^* \subseteq Z$, that are observable in the image $X$.

To achieve this, we propose concept selection by leveraging the Information Bottleneck (IB) principle, which seeks a compressed representation that preserves maximal information about another variable (Tishby et al., 2000). In our context, we need to find $Z^*$ that maximizes its mutual information with the image content $X$, i.e., $I(X; Z^*)$, while minimizing the redundancy by penalizing the mutual information between the selected concepts $Z^*$ and the full concept set $Z$, i.e., $I(Z^*; Z)$.

Formally, we define the optimization problem:

$$Z^* = \arg \max_{Z' \subseteq Z} \left[ I(X; Z') - \beta I(Z'; Z) \right], \tag{1}$$

where $\beta$ is a Lagrange multiplier that balances relevance and redundancy. However, directly computing $I(X; Z')$ is intractable due to the high dimensionality and complexity of the image space. To address this, we introduce an intermediate variable: a set of image descriptions $D = \{d_1, d_2, \ldots, d_n\}$, generated by prompting an LMM (i.e., the base LLaVA-1.5 (Liu et al., 2024a)), with instructions like *"Please describe the image."* Each description $d_i$ attempts to capture some aspects of the image content $X$. By increasing the number of collected descriptions $n$ with different prompts, we aim to approximate the true distribution of $X$.

This approach is analogous to assembling pieces of a puzzle: Each description provides partial information about the image, and collectively, they form a more complete representation. Similarly, multiple potentially partial descriptions generated from different prompts can collectively approximate the true image content. Under this intuition, we formalize the approximation in Theorem 1, with proof provided in the appendix.

**Theorem 1.** *Let $X$ be the true image content with label $c$ and $D = \{d_1, d_2, \ldots, d_n\}$ be independent and identically distributed (i.i.d.) samples from $P(D|X)$. Let $Z$ be an expert-defined concept list about label $c$. Under the assumptions of conditional independence and convergence (Assumptions 1 and 2), as $n \to \infty$, the mutual information $I(D; Z)$ converges to $I(X; Z)$:*

$$\lim_{n \to \infty} I(D; Z) = I(X; Z).$$

In practice, we cannot sample an infinite number of descriptions, and LMMs may generate inconsistent or contradictory descriptions due to hallucinations or uncertainties. To mitigate this, we design high-quality prompts and encourage diverse responses to improve the reliability of the generated descriptions. By doing so, we assume that most of the descriptions will accurately reflect the image content. This assumption is further validated through experiments in Section 4.4. From another perspective, if a feature is not consistently covered by image descriptions, it means that the model is not certain about its presence, which will naturally result in lower MI scores for the associated concepts, reducing their likelihood of being selected. Using the set of descriptions $D$, we reformulate the IB objective in Equation. 1 as:

$$Z^* = \arg \max_{Z' \subseteq Z} \left[ I(D; Z') - \beta I(Z'; Z) \right]. \tag{2}$$

However, computing mutual information in high-dimensional space directly remains challenging. Therefore, we employ the InfoNCE loss (Oord et al., 2018) as a lower-bound estimator of mutual information. For each concept $z_j \in Z$, we calculate an InfoNCE score $s_j$:

$$s_j = \sum_{d_i \in D} \log \frac{\exp \left( \text{sim} \left( \mathbf{e}_{d_i}, \mathbf{e}_{z_j} \right) / \tau \right)}{\exp \left( \text{sim} \left( \mathbf{e}_{d_i}, \mathbf{e}_{z_j} \right) / \tau \right) + \sum_{d_k \in \bar{D}} \exp \left( \text{sim} \left( \mathbf{e}_{d_k}, \mathbf{e}_{z_j} \right) / \tau \right)}, \tag{3}$$

where $\text{sim}(\cdot, \cdot)$ denotes the cosine similarity between embeddings, $\mathbf{e}_{d_i}$ and $\mathbf{e}_{z_j}$ are the embeddings of description $d_i$ and concept $z_j$, respectively, and $\tau$ is a temperature parameter. $\bar{D}$ are the descriptions for other images as negative samples. We can easily obtain the above embeddings through an off-the-shelf language embedding model (e.g., BERT (Devlin, 2018)). Next, we approximate $I(D; Z^*)$ as the sum of InfoNCE scores for the selected concepts: $I(D; Z^*) \approx \sum_{z_j \in Z^*} s_j$. To minimize $I(Z^*; Z)$, we selectively include concepts to reduce redundancy. Given that $Z^*$ is a subset of $Z$, we have: $H(Z^*) = I(Z^*; Z) = -\sum_{z_i \in Z^*} p(z_i) \log p(z_i)$, where $H(Z^*)$ is the entropy of $Z^*$. A selected subset $Z^*$ with a smaller size and higher probabilities for its $z_i$ elements will result in lower entropy. Combining these approximations, our selection criterion for $Z^*$ becomes:

$$Z^* = \left\{ z_j \in Z \,\middle|\, s_j > \mu + \hat{\beta}\sigma \right\}, \tag{4}$$

where $\mu$ and $\sigma$ are the mean and standard deviation of the InfoNCE scores $\{s_j\}_{j=1}^{|Z|}$, respectively. The parameter $\hat{\beta}$ controls the trade-off between including relevant concepts and avoiding redundancy. The selected concepts $Z^*$ are not only relevant to the image but also capture the most informative features unique to the class label, providing strong evidence for the classification result and serving as reasonable explanations. Once we obtain $Z^*$, we can **generate an explainable answer** for a classification query on the image $X$. Specifically, we prompt the base LMM with these concepts to produce a coherent explanation. The prompts used are detailed in the appendix. After gathering the image and query-answer pairs, we can use them to **fine-tune our LMMs**.

### 3.3 Step 2: Reward Model-Free Rejection Sampling

After the initial round of fine-tuning with explainable visual query-answer pairs, the LMMs have significantly improved their ability to generate reasonable explanations. This improvement allows us to leverage the current fine-tuned model to generate new data for subsequent training rounds. However, the quality of newly generated data can vary considerably in terms of label accuracy and explanation quality. Training on low-quality data could lead to performance degradation. To address this issue, we propose a rejection sampling technique that filters out low-quality outputs.

Rejection sampling, also known as Best-of-N, is an inference-time strategy that generates multiple candidates and selects the best one for further tuning (Touvron et al., 2023; Stiennon et al., 2020). In our work, we adapt this idea for visual fine-tuning to iteratively improve LMM performance. Our proposed rejection sampling process begins by generating a series of answer candidates, using the fine-tuned model $f_\theta^T$ from the last round $T$, and then identify the best answer from these candidates. For language-only conversations, this selection is typically performed by a reward model, which assigns higher rewards to answers aligning with desired criteria (Touvron et al., 2023). The answer with the highest reward is then selected and used for the next round ($T + 1$) of training. However, in the visual domain, finding a reliable reward model remains challenging, as noted by (Chen et al., 2024a). To address this, we propose a *reward model-free* data filtering method to select the highest quality interpretable answers.

Specifically, we leverage the selected concept set $Z^*$ from Section 3.2 as a reference to evaluate explanation quality. Our aim is to select the answer that best aligns with these relevant concepts. Formally, let $Y = \{y_1, y_2, \ldots, y_m\}$ represent the set of answers generated by the fine-tuned model for a given image. These answers are obtained by prompting the model with questions like *"What is the {item} in this image? Please provide your reasoning."* The *"item"* here is set to be an coarse-level label, like *bird, airplane*. Our goal is to select the answer $y^* \in Y$ that maximizes the mutual information with $Z^*$:

$$y^* = \arg\max_{y_i \in Y} I(y_i; Z^*). \tag{5}$$

We approximate $I(y_i; Z^*)$ using the InfoNCE score:

$$s_i' = \sum_{z_j \in Z^*} \log \frac{\exp\left(\text{sim}\left(\mathbf{e}_{y_i}, \mathbf{e}_{z_j}\right)/\tau\right)}{\exp\left(\text{sim}\left(\mathbf{e}_{y_i}, \mathbf{e}_{z_j}\right)/\tau\right) + \sum_{z_k \in Z, z_k \notin Z^*} \exp\left(\text{sim}\left(\mathbf{e}_{y_i}, \mathbf{e}_{z_k}\right)/\tau\right)}, \tag{6}$$

where $\mathbf{e}_{y_i}$ is the embedding of answer $y_i$. We select the answer with the highest score: $y^* = \arg\max_{y_i \in Y} s_i'$. The InfoNCE score provides a quantitative measure of how well the generated answer aligns with the relevant concepts. Therefore, our method eliminates the need for a separate reward model, which is particularly beneficial given the lack of reliable reward models for multi-modal data (Chen et al., 2024a). Additionally, we add another empirical constraint: the selected answer should contain the correct label $c$; otherwise, it will be discarded.

In summary, our framework, outlined in Algorithm 1, enhances the model's fine-grained classification capabilities by iteratively fine-tuning on diverse, high-quality, synthetic visual classification query-answer pairs. Step 1 identifies the most informative concepts in the image, while Step 2 selects the explanations that best align with these concepts. This two-step approach improves both the accuracy and interpretability of the model's predictions, enabling it to perform more effectively in complex visual classification tasks.

### 3.4 Theoretical Justification

Our approach is grounded in information theory, specifically the maximization of mutual information among the image $X$, the expert-defined concepts $Z$ for the whole class, and the generated answer $Y$ with explanations. Theoretically, the problem of finding the best $Y^*$ is equivalent to maximizing the three-way mutual information:

$$\max_{Y^*} I(X; Y; Z), \tag{7}$$

where $Y^*$ captures the maximum amount of information from both the image and the expert-defined concepts. However, directly optimizing this objective is intractable due to the high dimensionality of $X$ and $Y$ (Poole et al., 2019). To make the problem manageable, we decompose it into two subproblems corresponding to our two-step method.

In Step 1, we select a subset of concepts $Z^* \subseteq Z$ that maximizes $I(X; Z^*)$, capturing the most relevant concepts for the image. This aligns with the Information Bottleneck principle, where $Z^*$ serves as a distilled subset of $Z$ that retains maximal information about $X$. In Step 2, we select the generated explanation $Y^*$ that maximizes $I(Y^*; Z^*)$, ensuring that the generated explanation closely aligns with the selected concepts. With this decomposition, we establish a lower bound on the three-way mutual information (proof provided in appendix).

**Theorem 2.** *Let $X$, $Y$, and $Z$ be discrete random variables. Define:*

$$Z^* = \arg\max_{Z' \subseteq Z} I(X; Z') - \beta I(Z'; Z)$$
$$Y^* = \arg\max_{Y' \subseteq Y} I(Y'; Z^*)$$

*Then, the following inequality holds:*

$$I(X; Y; Z) \geq$$
$$I(Y^*; Z^*) + I(X; Z^*) - I(Z^*; Z).$$

This theoretical foundation justifies our two-step optimization process, ensuring that our method effectively captures the essential information shared among the image, concepts, and explanations.

---

**Algorithm 1** Iterative synthesis and fine-tuning.

**Require:** Dataset $\mathcal{D} = \{(X_i, c_i)\}_{i=1}^N$, Concept Sets $\mathcal{Z} = \{Z_i\}_{i=1}^{|C|}$ of every label, Pretrained LMM $f_\theta^0$, synthetic query set $Q = \{q_i\}_{i=1}^N$, Descriptive query $q_d$
**Ensure:** Fine-tuned LMM $f_\theta$ for accurate, interpretable explanations
1: Initialize training set $\mathcal{D}^* \leftarrow \emptyset$
2: **for** each $(X_i, c_i) \in \mathcal{D}$ **do**
3:     $Z \leftarrow \mathcal{Z}[c_i]$
4:     $D_i \leftarrow f_\theta^0(X_i, q_d)$
5:     **for** $z_j \in Z$ **do**
6:         $s_{ij} \leftarrow \text{InfoNCE}(D_i, z_j)$
7:     **end for**
8:     $\mu_i \leftarrow \text{mean}(\{s_{ij}\}), \sigma_i \leftarrow \text{std}(\{s_{ij}\})$
9:     $Z_i^* \leftarrow \{z_j \in Z \mid s_{ij} > \mu_i + \beta\sigma_i\}$
10:     $y_i \leftarrow \text{assemble}(Z_i^*)$
11:     $\mathcal{D}^* \leftarrow \mathcal{D}^* \cup \{(X_i, q_i, y_i)\}$
12: **end for**
13: **FT:** Fine-tune $f_\theta^0$ on $\mathcal{D}^*$ to obtain $f_\theta^1$
14: **repeat**
15:     **for** each $(X_i, \_) \in \mathcal{D}$ **do**
16:         $Y_i \leftarrow f_\theta^T(X_i, q_i)$
17:         **for** $y_{ij} \in Y_i$ **do**
18:             $s'_{ij} \leftarrow \text{InfoNCE}(Z_i^*, y_{ij})$
19:         **end for**
20:         $y_i^* \leftarrow \arg\max_{y_{ij}} s'_{ij}$
21:         $\mathcal{D}^* \leftarrow \mathcal{D}^* \cup \{(X_i, q_i, y_i^*)\}$
22:     **end for**
23:     **Iterative FT:** Fine-tune $f_\theta^T$ on $\mathcal{D}^*$
24: **until** Max iterations $T^m$
25: **return** $f_\theta^m$

---

## 4 EXPERIMENTS

We conduct experiments to address the following questions. **Q1**: How effectively does our method improve performance on fine-grained visual classification tasks? **Q2**: How does our method enhance the explainability of the fine-tuned model? **Q3**: How does our concept selection strategy compare to baseline methods? **Q4**: Does our training scheme mitigate shortcut learning? **Q5**: How usable is our trained model?

### 4.1 EXPERIMENTS SETTINGS

**Implementation Details:** We evaluated our approach on a variety of datasets, including fine-grained classification datasets (CUB-200 (Wah et al., 2011), Stanford Dogs (Khosla et al., 2011), FGVC-Aircraft (Maji et al., 2013)), medical datasets (HAM10000 (Tschandl et al., 2018), Chest X-Ray for Pneumonia (Kermany et al., 2018)), and the Plant Disease Dataset (PLD) (Vipooool, 2020). By testing on datasets from diverse domains, we demonstrate the versatility of our proposed framework with self-synthesized data. Our experiments used LLava-1.5-7B (Liu et al., 2024a) as the base LMM, and for the Chest X-Ray datasets, we employed its medical version (Li et al., 2024b). Additionally, E5 (Wang et al., 2022) served as the embedding model. We fine-tuned the LMMs using LoRA, focusing on all linear layers. Training was conducted on 8 H100 GPUs, utilizing Hugging-Face (Wolf, 2019) and DeepSpeed frameworks for efficient distributed training and optimization. Further details are provided in the appendix. We also provide our code here[1].

**Baselines:** Given the novel problem of generating interpretable answers without image-specific annotations, we design the following baselines for comparisons: (1) **Base LMM**: Assesses the base multimodal model's performance in the zero-shot setting. (2) **Naive Label Fine-tuning (NL)**:

---

[1] https://github.com/sycny/SelfSynthX

Table 1: Our method achieves superior accuracy and explanation quality across diverse datasets.

| Dataset | Method | Accuracy ↑ Per Iteration | | | | Explanation Quality | | | General Ability |
|---|---|---|---|---|---|---|---|---|---|
| | | 1 | 2 | 3 | 4 | EE ↑ | CS ↑ | FS ↓ | MMMU ↑ |
| CUB-200 | Base | 2.69 | – | – | – | 0.92 | 0.67 | 4.28 | 35.56 |
| | NL | 73.42 | 78.25 | 79.94 | 82.21 | 0.00 | – | – | 35.67 |
| | L+GE | 61.48 | 72.23 | 73.23 | 73.06 | 1.00 | 0.70 | 6.84 | 34.89 |
| | Ours | 80.24 | 83.76 | 84.69 | 85.02 | 1.00 | 0.82 | 6.53 | 35.00 |
| Stanford dogs | Base | 12.2 | – | – | – | 0.94 | 0.69 | 5.47 | 35.56 |
| | NL | 82.73 | 82.34 | 84.03 | 84.27 | 0.00 | – | – | 34.67 |
| | L+GE | 73.45 | 77.89 | 78.15 | 76.55 | 1.00 | 0.77 | 7.50 | 34.56 |
| | Ours | 85.29 | 86.75 | 86.86 | 86.91 | 1.00 | 0.86 | 7.41 | 34.56 |
| FGVC-A | Base | 3.00 | – | – | – | 0.97 | 0.42 | 5.39 | 35.56 |
| | NL | 83.47 | 87.28 | 87.82 | 87.73 | 0.00 | – | – | 35.56 |
| | L+GE | 72.13 | 79.87 | 82.45 | 82.69 | 1.0 | 0.76 | 8.59 | 35.56 |
| | Ours | 88.78 | 90.91 | 91.42 | 91.99 | 1.0 | 0.79 | 7.00 | 37.33 |
| PLD | Base | 0.00 | – | – | – | 0.95 | – | – | 35.56 |
| | NL | 89.38 | 94.52 | 94.29 | 93.95 | 0.00 | – | – | 34.78 |
| | L+GE | 24.03 | 25.27 | 24.56 | 24.90 | 1.00 | 0.76 | 10.45 | 35.44 |
| | Ours | 75.96 | 92.80 | 96.59 | 97.16 | 1.00 | 0.86 | 9.01 | 35.22 |
| HAM10000 | Base | 1.62 | – | – | – | 0.98 | 0.63 | 3.93 | 35.56 |
| | NL | 77.28 | 80.75 | 82.49 | 81.71 | 0.00 | – | – | 35.33 |
| | L+GE | 7.47 | 8.83 | 9.35 | 8.45 | 1.00 | 0.94 | 9.68 | 35.22 |
| | Ours | 79.37 | 82.29 | 83.69 | 85.06 | 1.00 | 0.87 | 7.43 | 35.89 |
| Chest X-ray Pneumonia (LLaVA-Med) | Base | 62.50 | – | – | – | 1.00 | 0.24 | 3.49 | – |
| | NL | 85.58 | 89.10 | 85.90 | 89.58 | 0.00 | – | – | – |
| | L+GE | 62.50 | 62.50 | 62.98 | 62.66 | 1.00 | 0.79 | 7.19 | – |
| | Ours | 97.60 | 96.31 | 99.04 | 98.72 | 1.00 | 0.87 | 8.25 | – |

*Base: original model; NL: only train with labels; L+GE: train with labels and general explanations*

Fine-tunes the base model using only class labels and a simple template (e.g., *"This is a picture of {label}"*). (3) **Label with General Explanations (L+GE)**: We adopt the data synthesis approach introduced in the LLaVA paper (Liu et al., 2024b) and (Kim & Ji, 2024), where training data is generated using a language-only model (e.g., GPT-4), based on class labels and their corresponding label-level knowledge. More details, including the prompts, are in the appendix.

## 4.2 RQ1: TRAINING ON OUR SYNTHETIC DATA IMPROVES CLASSIFICATION

We evaluated our model's classification capabilities using a multi-round, progressive rejection sampling training process comprising four iterations, each with two epochs. Classification accuracy was measured after each iteration, with success defined as the presence of the ground truth label in the model's response (Kim & Ji, 2024). All trainable baselines were trained and evaluated under identical settings to ours for a fair comparison. More details on training are provided in the appendix.

Our proposed method achieves higher accuracy than both NL and L+GE baselines, as shown in Table 1. While the baselines, particularly NL, exhibit overfitting, our method improves accuracy with each iteration, demonstrating its effectiveness in fine-grained classification tasks. Our approach's resilience to overfitting stems from the use of rejection sampling, which generates a more diverse training dataset. This expanded data pool enhances the model's generalization capabilities. Moreover, training with our dataset, which includes specific visual features, helps the model learn more detailed visual knowledge, contributing to further gains in classification performance. In contrast, the L+GE baseline struggles with certain datasets, notably HAM10000, likely due to irrelevant information in general explanations impeding effective learning. Moreover, we assessed the general ability of the models using the MMMU metric (Yue et al., 2024) evaluated by the LMMs-eval tool (Zhang et al., 2024). The results indicate that all trained models maintain comparable general abilities to the base model, with negligible degradation. Note that the LMMs-eval tool does not support the LLaVA-Med model; therefore, the general ability metric is not reported. More experiment results can be found in the appendix.

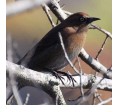 **Ours:** The specific name of this bird is the **Rusty Blackbird.** The Rusty Blackbird is a medium-sized blackbird with a slightly curved, slender black bill. It is known for its rusty or brownish tones on its body, particularly in the non-breeding season. The bird is often seen in large flocks and is commonly found in wetland habitats.

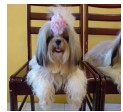 **Ours:** "The specific name of this dog is a **Tzu.** The Tzu is a small, compact dog with a square-like frame, short legs, and a short muzzle. The dog's hair is often tied up with a bow on top of the head, and it has a distinctive undershot bite."

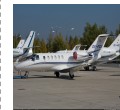 **Ours:** The airplane shown in this image is a **Cessna Citation.** This conclusion is based on the visual features of the aircraft, such as the short, rounded wing roots connecting to the fuselage, the low-wing design, the distinctive nose with a slight downward curve, and the retractable tricycle landing gear. These features are characteristic of Cessna Citation aircraft, which are known for their performance, comfort, and reliability.

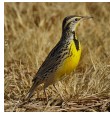 **Ours:** The specific name of this bird is the **Western Meadowlark.** The Western Meadowlark is a small bird with a bright yellow underbelly, a black \"V\" shape on the chest, and a long, pointed bill. It is commonly found in the grasslands and meadows of the western United States.

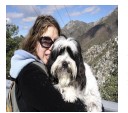 **Ours:** "The specific name of this dog is a **Tibetan Terrier.** This breed is known for its distinctive appearance, which includes a wide range of colors, a pronounced beard and moustache, and a square proportioned body. The dog in the image has a black and white coat, which is a common color pattern for this breed."

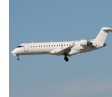 **Ours:** The airplane shown in this image is a **Canadair CRJ-900.** This conclusion is based on the following visual features:\n\n1. The presence of two engines mounted on the rear fuselage, which is a characteristic feature of the CRJ-900.\n2. The slanted cockpit windshield with a modern, streamlined appearance, which is consistent with the design of the CRJ-900.\n3. The long, narrow fuselage with a sleek, aerodynamic design, which is also a common feature of the CRJ-900.

Figure 4: Our generated answers contain detailed visual explanations.

### 4.3 RQ2: OUR METHOD PROVIDES HIGH-QUALITY EXPLANATION

Assessing the quality of generated explanations is challenging, especially without case-by-case ground truth annotations (Ding et al., 2022; Schuff et al., 2022). Following Mohseni et al. (2021), we evaluate our explanation from three aspects: explanation existence, cognition level, and fluency.

**Explanation Existence (EE).** This metric assesses a model's capability to generate explanations (Xu et al., 2023). Following Bills et al. (2023); Bricken et al. (2023), we employ an advanced proprietary LLM (GPT-4o) to determine whether each model-generated answer includes an explanation. Formally, for a set of generated answers $Y = \{y_1, y_2, \ldots, y_n\}$ queried by prompt like "*What is the bird name? Provide your reason.*", EE is defined as: $\text{EE} = \frac{1}{n} \sum_{i=1}^{n} e_i$, where $e_i = 1$ if the $i^{\text{th}}$ answer $y_i$ includes an explanation, and $e_i = 0$ otherwise. We provide the evaluation prompt in the appendix.

Results: As shown in Table 1, our method achieves an EE of 1.00 across all datasets, indicating that it consistently produces explanations for its predictions. The base model can generate explanations at most times but fails to do so in some cases. In contrast, the NL baseline records an EE of 0.00, reflecting its inability to generate explanations due to training solely on class labels. The L+GE method also attains an EE of 1.00 but falls short in other quality metrics compared to our approach.

**Cognition Score (CS).** This metric evaluates the coherence and logical flow of generated explanations (Nourani et al., 2019; Fan et al., 2020). Following Liu et al. (2024e); Bills et al. (2023); Lieberum et al. (2024), we employ an advanced proprietary LLM (GPT-4o) to assess the rational integrity of explanations by analyzing their alignment with expert knowledge. For an answer $y_i$ containing a label and explanation, we first extract label-level concepts $Z$ corresponding to the label. We then use an evaluation prompt to obtain a cognition score from LLM: $cs_i = \text{LLM}(y_i, Z, \text{Eval\_Prompt})$. The Eval_Prompt is detailed in the appendix. Scores range from 0 to 1, with higher scores indicating better alignment with expert-predefined knowledge. We only assess correct answers. The total CS is calculated as: $\text{CS} = \frac{1}{|Y_c|} \sum_{i=1}^{|Y_c|} cs_i$, where $Y_c$ is the set of correct answers.

Results: Our method generally achieves higher CS values than the baselines, demonstrating superior coherence and logical reasoning in the explanations. This improvement reflects the effectiveness of our approach in generating explanations that are more useful and trustworthy to human users. The reason behind our better cognition is that we select image-level concepts to build the training dataset, rather than using all label-level features. This ensures that our model focuses on relevant features present in the image and effectively connects the image content with expert-defined concepts, avoiding distractions from irrelevant features not shown in the image.

**Fluency Score (FS).** This metric evaluates the grammatical correctness and naturalness of the explanations (Liu et al., 2021). We use perplexity, a standard metric that measures how well a language model predicts the text, with lower values indicating better fluency.

Results: Our method demonstrates good fluency in the generated explanations. The base model receives the best fluency. This is because training on expert-level knowledge inevitably increases the complexity of explanations, which can lead to higher perplexity scores. However, our method generally achieves lower perplexity than the L+GE baseline, indicating that our explanations are more natural, highlighting the robustness of our approach.

In summary, our proposed method not only enhances classification accuracy but also consistently generates high-quality explanations that are coherent, logical, and fluent. We also show some quantitative results in our explanation in Figure 4. More results can be found in the appendix.

## 4.4 RQ3: Evaluation on Interpretable Concepts Selection

To evaluate the efficacy of our proposed model in select-
ing expert-defined features from images, we conducted a
comprehensive study with human experts. Experts were
invited to identify and annotate the top-4 relevant con-
cepts for each image from a dataset of six bird species,
providing ground truth annotations for our evaluation.
Our investigation examined two primary aspects: the ef-
fect of varying the number of descriptions on concept se-
lection performance, and a comparative analysis against
several baseline methods. These baselines include GPT-
4o for concept extraction, which relied on a carefully
crafted prompt to guide the model in identifying and re-
turning the four most probable concepts for each image.
We applied the same approach to the LLaVA model for
concept extraction. In contrast, for the CLIP model, con-
cept selection was performed by identifying top-4 con-
cepts with the highest CLIPScore (Hessel et al., 2021)
relative to the target image.

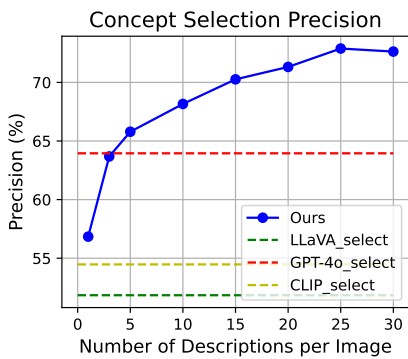

Figure 5: Our method demonstrates superior precision in concept selection compared to applying GPT-4o.

We evaluated performance using precision, comparing model-selected concepts with human-
annotated ground truth. Results demonstrated that our model outperforms strong baselines in the
concept selection task. Its precision improves proportionally with the increased number of descrip-
tions, peaking at 72.89% with 25 descriptions. In contrast, GPT-4o maintained a relatively high
precision of 63.95%, while both LLaVA and CLIP showed weaker performances at approximately
55%. These findings highlight our model's superior ability to leverage repeated sampled descrip-
tions for more accurate concept selection. We show some qualitative results in Figure 6, with more
in the appendix.

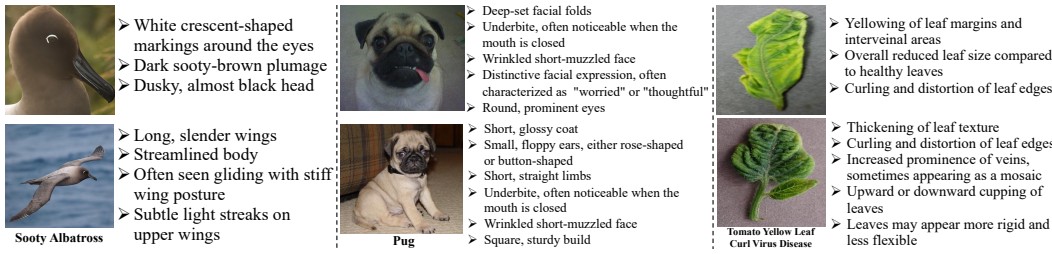

Figure 6: Different image-level visual concepts for objects with the same label.

## 4.5 RQ4: Visualization of Visual Focus of LMMs

To interpret our model's predictions, we visualize the outputs of LMMs using a gradient-based ex-
planation method (Wu et al., 2023). This approach identifies the image regions most influential in
generating the model's answer by computing gradients of answer token probabilities with respect
to image pixels. Figure 7 presents case examples demonstrating that training on our synthetic an-
swers with cognitive explanations can effectively prevent the model from learning spurious features,
whereas directly linking images to naive labels may result in shortcut learning.

## 4.6 RQ5: Evaluation on Usability

Previous LMMs often struggle with specialized questions due to difficulty recognizing domain-
specific objects. Our method enhances LMMs' ability to identify specialized objects and provide
verifiable visual details, as demonstrated in Table 10. In Case 1, the LMM accurately identifies a bell
pepper leaf disease, describing specific visual cues such as brown necrotic spots and chlorosis. It
then recommends an appropriate treatment. In Case 2, the LMM correctly identifies a male Northern
Cardinal, noting distinctive features like its bright red plumage and black mask. These improvements
enable LMMs to serve as **effective visual assistants** in specialized domains, offering users accurate
and interpretable responses that can be verified against the provided images.

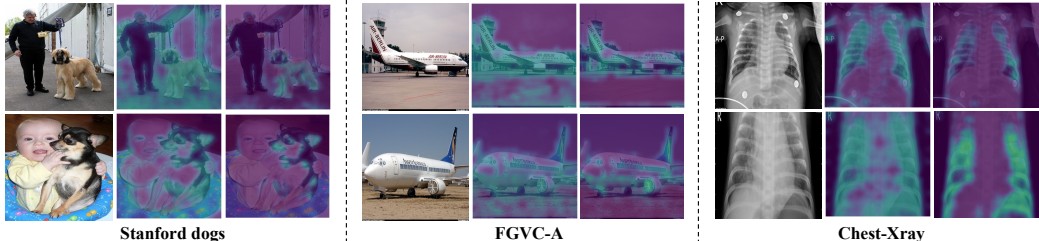

Figure 7: Visualization of heatmaps for different datasets. From left to right in each triplet: **Original Image**, **Heatmap of NL model**, **Heatmap of Ours**.

Table 2: LMMs fine-tuned with our method provide verifiable details, enhancing user trust.

| | | |
|---|---|---|
| | User: | What is the name of this leaf disease? Can you give me your reason? |
| | LMM: | The name of this leaf disease is **Bell Pepper Leaf with Bacterial Spot Disease**. The reason for this name is that the leaf has *brown or black necrotic spots*, which are characteristic of bacterial infections. These spots can coalesce, forming *larger necrotic areas*, and the leaf may exhibit *chlorosis or yellowing* of surrounding leaf tissue. The spots are often located on *the underside of the leaf*, and the leaf may become *crinkled or deformed* near affected areas. |
| | User: | OK then, what chemical should I use to treat this disease? |
| | LMM: | It is recommended to use a **copper-based fungicide** to treat this disease. |
| | User: | Identify this bird. What features led to your conclusion? |
| | LMM: | The bird is a **male Northern Cardinal** with a *bright red plumage* on its head, throat, and chest. The presence of *a crest on its head* and a *black "mask" around its eyes* further supports the identification of this bird. |
| | User: | Where can I find this bird? |
| | LMM: | You can find this bird in **the United States**, specifically in the **eastern and central parts** of the country. The Northern Cardinal is a common resident bird in areas with dense woodlands, shrubs, and gardens. |

## 5 RELATED WORK: SELF-IMPROVEMENT AND DATA SYNTHESIS

In the field of LLMs, self-generated data has become a powerful tool for enhancing model performance (Liu et al., 2024d). Researchers have explored various techniques, such as rejection sampling (Touvron et al., 2023; Dubey et al., 2024), self-rewarding (Yuan et al., 2024), and self-play (Chen et al., 2024c), enabling models to improve using synthetic data. Recent studies (Snell et al., 2024; Brown et al., 2024) have proposed the "inference-time scaling law", suggesting that increasing inference samples size boosts the likelihood of generating high-quality data. Data synthesis techniques have also been applied to LMMs, improving general vision tasks like visual question answering (Zhou et al., 2024a; Deng et al., 2024; Gao et al., 2023; Zhou et al., 2024b) and enhancing instruction-following capabilities (Luo et al., 2024). Our work extends this line of research by focusing on domain-specific visual classification to enable effective visual assistance in professional tasks. In contrast to existing methods, we address the unique challenges of data synthesis in specialized domains, thereby extending these techniques to support expert-driven applications.

## 6 CONCLUSION

In this work, we addressed LMMs' limitations in domain-specific visual classification tasks by introducing a novel framework that enhances their cognition and explainability through iterative fine-tuning with self-synthesized interpretable answers. By applying the IB principle to select relevant visual concepts without extensive image-specific annotations, our approach significantly improved classification accuracy and explanation quality across various datasets. By enabling LMMs to generate accurate, interpretable explanations grounded in domain-specific visual features, our framework advances their applicability in specialized domains, paving the way for more reliable multimodal models in knowledge-intensive applications. Future work may explore more complex tasks than classification tasks, and refinements to enhance scalability and generalizability.

ACKNOWLEDGMENTS

The work is, in part, supported by NSF (#IIS-2223768). Dr. Xiang Li is supported by the Google Research Scholar Program. The views and conclusions in this paper are those of the authors and should not be interpreted as representing any funding agencies.

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

## A  THEORETICAL PROOF ON THEOREM 1

**Theorem 3.** *Let $X$ be the true image content with label $c$ and $D = \{d_1, d_2, \ldots, d_n\}$ be independent and identically distributed (i.i.d.) samples from $P(D|X)$. Let $Z$ be an expert-defined concept list about label $c$. Under the assumptions of conditional independence and convergence (Assumptions 1 and 2), as $n \to \infty$, the mutual information $I(D; Z)$ converges to $I(X; Z)$:*

$$\lim_{n \to \infty} I(D; Z) = I(X; Z).$$

*Proof.* We proceed with the following steps:

### A.1  ASSUMPTIONS

**Assumption 1.** *Conditional Independence: $d_1, d_2, \ldots, d_n$ are conditionally independent of $Z$ given $X$.*

**Assumption 2.** *Convergence of Mutual Information: As $n \to \infty$, the mutual information between $D$ and $X$ converges to the entropy of $X$:*

$$\lim_{n \to \infty} I(D; X) = H(X)$$

### A.2  PROOF STEPS

1. Mutual Information Expression: We start with the definition of mutual information:
$$I(D; Z) = H(D) - H(D|Z)$$

2. Expanding $H(D|Z)$ Using the Chain Rule: Apply the chain rule for entropy:
$$H(D|Z) = H(D|Z, X) + H(X|Z) - H(X|D, Z)$$
Substituting this back into the mutual information expression:
$$I(D; Z) = H(D) - [H(D|Z, X) + H(X|Z) - H(X|D, Z)]$$

3. Applying Conditional Independence: By Assumption 1, we have:
$$H(D|Z, X) = H(D|X)$$
Therefore,
$$I(D; Z) = H(D) - H(D|X) - H(X|Z) + H(X|D, Z)$$
Recognizing $I(D; X) = H(D) - H(D|X)$, we have:
$$I(D; Z) = I(D; X) - H(X|Z) + H(X|D, Z)$$

4. Taking the Limit as $n \to \infty$: Apply the limit to both sides:
$$\lim_{n \to \infty} I(D; Z) = \lim_{n \to \infty} [I(D; X) - H(X|Z) + H(X|D, Z)]$$

   By Assumption 2, we have:
$$\lim_{n \to \infty} I(D; X) = H(X)$$

   For the term $H(X|D, Z)$, we argue that:
$$\lim_{n \to \infty} H(X|D, Z) = 0$$
This follows from Assumption 2, as it implies that $D$ becomes a sufficient statistic for $X$ as $n \to \infty$. Therefore, conditioning on $Z$ does not add any information about $X$ once we have $D$.

5. Conclusion: Substituting these limits into our equation:
$$\lim_{n \to \infty} I(D; Z) = H(X) - H(X|Z) + 0 = I(X; Z)$$

This demonstrates that as the number of sampled descriptions $n$ increases indefinitely, the mutual information between the aggregated descriptions $D$ and the human concepts $Z$ converges to the mutual information between the true image content $X$ and $Z$. $\square$

### A.3 ADDITIONAL NOTES

#### A.3.1 CONVERGENCE OF $I(D; X)$ TO $H(X)$

The statement:
$$\lim_{n \to \infty} I(D; X) = H(X)$$

This is directly from Assumption 2. To understand its implications:

- Recall that mutual information is defined as: $I(D; X) = H(X) - H(X|D)$
- For this equality to hold as $n \to \infty$, it must be true that:
$$\lim_{n \to \infty} H(X|D) = 0$$
- This means that as we gather more samples ($D$), we eliminate all uncertainty about $X$.
- In other words, with infinite samples, $D$ contains all information about $X$.

#### A.3.2 CONVERGENCE OF $H(X|D, Z)$ TO 0

The statement:
$$\lim_{n \to \infty} H(X|D, Z) = 0$$

This follows from the previous point. Here's the reasoning:

- We've established that as $n \to \infty$, $D$ contains all information about $X$.
- This means $D$ becomes a *sufficient statistic* for $X$.
- A sufficient statistic contains all the information that the sample provides about the parameter (in this case, $X$).
- Therefore, once we know $D$, knowing $Z$ doesn't provide any additional information about $X$.
- Mathematically, this means: $H(X|D, Z) = H(X|D)$
- But we know from the first part that $\lim_{n \to \infty} H(X|D) = 0$
- Thus, $\lim_{n \to \infty} H(X|D, Z) = 0$

## B THEORETICAL PROOF ON THEOREM 2

**Theorem 4.** *Let $X$, $Y$, and $Z$ be discrete random variables. Define:*
$$Z^* = \arg\max_{Z' \subseteq Z} I(X; Z') - \beta I(Z'; Z)$$
$$Y^* = \arg\max_{Y' \subseteq Y} I(Y'; Z^*)$$

*Then, the following inequality holds:*
$$I(X; Y; Z) \geq I(Y^*; Z^*) + I(X; Z^*) - I(Z^*; Z). \tag{8}$$

*where $I(\cdot; \cdot)$ denotes mutual information and $H(\cdot)$ denotes entropy.*

*Proof.* We prove this theorem using fundamental principles of information theory:

1. Recall the definition of multivariate mutual information:
$$I(X; Y; Z) = I(X; Y) - I(X; Y|Z) \tag{9}$$

2. By the chain rule of mutual information (Shi et al., 2023), we can rewrite this as:
$$I(X; Y; Z) = I(X; Z) + I(Y; Z) - I(X, Y; Z) \tag{10}$$

3. Consider our subsets $Z^*$ and $Y^*$. By definition of mutual information and the data processing inequality:

$$I(Y^*; Z^*) \leq I(Y; Z^*) \tag{11}$$
$$I(X; Z^*) \leq I(X; Z) \tag{12}$$

4. Substituting these into our equation:

$$I(X; Y; Z) \geq I(X; Z^*) + I(Y^*; Z^*) - I(X, Y; Z) \tag{13}$$

5. For any random variables $A$ and $B$:

$$I(A; B) \leq \min(H(A), H(B)) \tag{14}$$

Therefore:
$$I(X, Y; Z) \leq \min(H(X, Y), H(Z)) \leq I(Z^*; Z) \tag{15}$$

6. Applying this to our inequality:

$$I(X; Y; Z) \geq I(X; Z^*) + I(Y^*; Z^*) - I(Z^*; Z) \tag{16}$$

7. Therefore, we can conclude:

$$I(X; Y; Z) \geq I(Y^*; Z^*) + I(X; Z^*) - I(Z^*; Z) \tag{17}$$

$\square$

# C    PROMPTS FOR DIFFERENT TASKS

EXAMPLE PROMPT FOR BIRD IMAGE DESCRIPTION IN CUB-200

| Prompt 1 | Focus solely on the bird shown in the image. Describe the bird's appearance in detail, emphasizing its most prominent physical features. Avoid mentioning the background or other elements not related to the bird. |
|---|---|
| Prompt 2 | Provide a focused analysis of the bird in this image, detailing its distinctive physical features. Concentrate exclusively on the bird and describe its appearance without referencing the surroundings or any extraneous details. |
| Prompt 3 | Directly observe the bird depicted and offer a precise description of its visual attributes. Ensure your description is limited to the bird itself, detailing its primary features and omitting any unrelated background elements. |

Table 3: Prompts for bird image analysis in CUB-200 dataset

EXAMPLE REWRITE PROMPTS FOR DIFFERENT DATASETS

| Dataset | Prompt |
|---|---|
| cub-200 / stanford_dogs | This is a picture of a {label} with the following visual features: {concepts_str}. Based on the information provided, please answer the following question.
**Question:** '{query}' |
| HAM10000 | This is a dermatoscopic image of {label} disease with the following visual features: {concepts_str}. Based on the information provided, please answer the following question.
**Question:** '{query}' |
| PLD / fgvc | This is a picture of {label} with the following visual features: {concepts_str}. Based on the information provided, please answer the following question.
**Question:** '{query}' |
| chest-xray | This is a chest-xray of {label} with the following visual features: {concepts_str}. Based on the information provided, please answer the following question.
**Question:** '{query}' |

Table 4: Answer rewrite prompts for different datasets.

L+GE SYNTHESIZE PROMPT EXAMPLE

| Dataset | Prompt |
|---|---|
| L+GE synthesize prompt | There is a picture of a {label}, which is known for the following characteristics: {concepts_str}.
Act as if you can see the picture. Please answer the following question based on the above information. Make your answer concise.
**Question:** '{query}'
**Answer:** |

Table 5: Prompt for answering questions based on image characteristics for general image datasets.

EVALUATION PROMPTS FOR EXPLANATION EXISTENCE (EE) AND COGNITION SCORE (CS)

| Prompt Type | Prompt |
|---|---|
| **EE Prompt** | Determine whether the following answer contains a valid explanation supporting its conclusion. Respond with only 'true' or 'false'. 

 **Answer:** {answer} 

 Contains an explanation? |
| **CS Prompt** | Evaluate the coherence and logical alignment of the following explanation with the provided concepts. Please note: the explanation does not need to fully encompass all concepts. 

 Assign a consistency score between 0 and 1, where 1 indicates the explanation contains no irrelevant information to the listed concepts, and 0 indicates complete misalignment with entirely irrelevant information. Only give the score. 

 **Concepts:** {concepts_formatted} 
 **Explanation:** {explanation} 

 **Consistency Score:** |

Table 6: Prompts for Explanation Existence (EE) and Cognition Score (CS).

CONCEPT EXTRACTION PROMPTS

To obtain label-level concepts, we used GPT-4o with prompts designed to elicit detailed visual features associated with each class label. An example prompt is:

| Prompt Type | Prompt |
|---|---|
| **Concept Extraction Prompt** | Please provide a list of visual characteristics that are commonly associated with the bird species {Class Name}. Include features such as color patterns, shapes, and distinctive markings. |

Table 7: Prompts for Concept Extraction.

# D    EXPERIMENTS DETAILS

## D.1    TRAINING FRAMEWORK IMPLEMENTATION

Our experimental setup leverages state-of-the-art techniques for fine-tuning the LLaVA-1.5-7B model. We initialize the model using pre-trained weights obtained from the Hugging Face model hub[2]. Our training process uses the SFTTrainer for supervised fine-tuning and implement Low-Rank Adaptation (LoRA) to achieve efficient parameter fine-tuning. In our configuration, we set the LoRA rank ($r$) to 128, the scaling factor ($\alpha$) to 256, and the dropout rate to 0.1. We fine tune model on a mixture of our domain-specific synthetic data with LLaVA-1.5's original general instruction-following training data to stabilize training without introducing additional data. We enable gradient checkpointing to reduce memory usage, allowing us to process larger batch sizes by not storing all intermediate activations. In addition, we employ DeepSpeed ZeRO Stage 3 to fully utilize our multi-GPU configuration. We also utilize mixed precision training with the BF16 format, which further reduces memory requirements while maintaining computational performance. Gradient accumulation is implemented with 2 steps and a per-device train batch size of 16. Moreover, slight data augmentation are applied to the image data to create diversity and enhance training robustness.

## D.2    DATASET OVERVIEW AND EXPERIMENTAL RUNNING TIMES

Table 8: Summary of Selected Datasets

| Dataset Name | Test / Training Cases | Label Categories | Running Time Per Iter |
|---|---|---|---|
| CUB-200-2011 | 5,794 / 5,994 | 200 | 143 minutes |
| FGVC-Aircraft | 3,333 / 6,667 | 100 | 171 minutes |
| Stanford Dogs | 8,580 / 12,000 | 120 | 310 minutes |
| Chest X-ray images | 624 / 5,216 | 2 | 113 minutes |
| HAM1000 | 5,008 / 5,007 | 7 | 107 minutes |
| PLD | 47,523 / 5,279 | 37 | 109 minutes |

We use the original labels and train/test split from the original datasets. Detailed train/test data statics are reported in Table 8. Regarding the experiments running time, each iteration of rejection sampling and fine-tuning took approximately 2.5 hours on a single NVIDIA H100 GPU for datasets like CUB-200 and FGVC-Aircraft. A more detailed report can be found in Table 8.

# E    VISUALIZATION RESULTS

---

[2] https://huggingface.co/llava-hf/llava-1.5-7b-hf

Table 9: Generated Answer Examples.

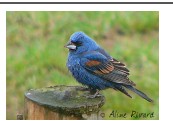

User: Based on the visual content, what is the specific name of this bird? Provide an explanation.

LMM: The specific name of this bird is the Yellow breasted Chat. This bird is known for its bright yellow throat and chest, olive-green back, wings, and tail, and short, rounded wings. It is a small bird that is commonly found in the Americas, particularly in the United States and Canada.

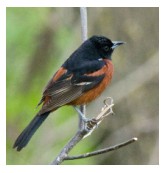

User: Based on the visual content, what is the specific name of this bird? Provide an explanation.

LMM: The specific name of this bird is the Blue Grosbeak. The Blue Grosbeak is a small, colorful bird with a deep blue overall plumage in males and a brown overall coloration with subtle blue tinges and faint streaking in females. The bird has a slightly glossy sheen to its blue feathers and is known for its melodious songs.

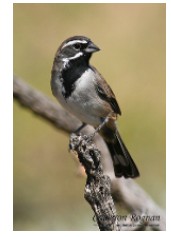

User: Based on the visual content, what is the specific name of this bird? Provide an explanation.

LMM: The specific name of this bird is the Orchard Oriole. The bird has a black head, back, throat, and upper chest, with a rich chestnut-brown underparts. It has a black tail and a black throat patch developing. The bird's wings are black with white wing bars, and it has a black bill. The Orchard Oriole is a small bird with a distinctive appearance, and it is known for its melodious songs.

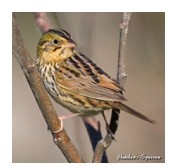

User: Based on the visual content, what is the specific name of this bird? Provide an explanation.

LMM: The specific name of this bird is the Black throated Sparrow. This bird is known for its distinctive black throat patch and white stripe below its eyes. It is a small bird with a brown and white coloration, and it is often seen perched on branches or low vegetation.

User: Based on the visual content, what is the specific name of this bird? Provide an explanation.

LMM: The specific name of this bird is the Henslow Sparrow. This bird is known for its olive-colored head, short, pointed bill, and pale eye-ring. It is a small bird with a brown and white coloration, and it is often seen flitting low in dense grassy habitats.

## F    ADDITIONAL EXPERIMENTS

### F.1    IMPACT ON GENERAL ABILITIES

To assess the effect of our proposed training method on the general abilities of the models, we conducted evaluations on four widely used multimodal benchmarks: MMStar, SEED-Bench-2 Plus, MMBench, and MME (Cognition). Table 11 summarizes the results.

- **MMStar** (Chen et al., 2024b): MMStar is a benchmark designed to evaluate vision-indispensable multimodal capabilities of MLLMs. It comprises 1,500 high-quality samples selected through a meticulous process from an initial pool of 22,401 samples. The benchmark assesses six core capabilities, each containing 250 samples, distributed across 18 detailed axes. This structure ensures a comprehensive evaluation of models' performance in tasks that require visual understanding.

- **SEED-Bench-2 Plus** (Li et al., 2024a): SEED-Bench-2 Plus is specifically designed to evaluate text-rich visual comprehension in MLLMs. It features 2,300 multiple-choice questions with precise human annotations, spanning three broad categories: Charts, Maps, and Webs. These categories encompass a wide spectrum of real-world text-rich scenarios, effectively simulating environments where visual and textual information are intertwined. The benchmark aims to assess models' proficiency in interpreting complex visual data embedded with textual content.

Table 10: Selected Concepts Examples.

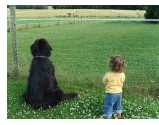 Concepts: "Powerful, thick neck"
"Broad and strong back"
"Large and muscular build"
"Well-sprung ribs"

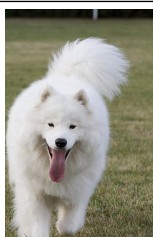 Concepts: "Pure white or cream-colored fur"
"Well-feathered tail that blends with the body fur"
"Strong, straight legs"
"Broad head with a slightly rounded skull"

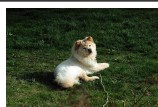 Concepts: "Straight, arched tail that rests on the back"
"Broad, flat skull"
"Thick double coat, either rough or smooth"

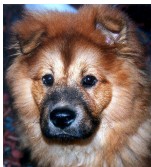 Concepts: "Deep-set, almond-shaped eyes"
"Erect, triangular ears"
"Thick ruff of fur around the neck"
"Thick double coat, either rough or smooth"

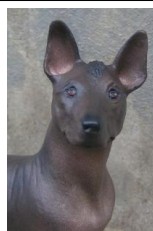 Concepts: "Ears: Large, bat-like ears that stand erect and move attentively."
"Distinctive facial features: Elongated muzzle with a moderate stop and expressive eyes."
"Whiskers: Short or absent due to the lack of hair follicles around the muzzle area."

- **MMBench** (Liu et al., 2024e): MMBench is a comprehensive benchmark that evaluates MLLMs across 20 ability dimensions. It includes approximately 3,000 multiple-choice questions, each with a single correct answer. MMBench addresses limitations of traditional benchmarks by focusing on fine-grained abilities and introducing robust evaluation strategies. The benchmark employs ChatGPT to match a model's prediction with the choices of a question, providing a more reliable assessment of model performance.

- **MME (Cognition)** (Fu et al., 2023): MME (Cognition) benchmark evaluates the cognitive abilities of MLLMs through tasks requiring reasoning and understanding of visual and textual inputs. The cognition part includes four subtasks: Commonsense Reasoning, Numerical Calculation, Text Translation, and Code Reasoning. These tasks test the model's ability to integrate multimodal information, such as interpreting visual scenes, performing arithmetic based on images, translating text in images, and reasoning about code snippets. The dataset consists of carefully curated images and instruction-answer pairs to ensure robust and fair assessment.

As shown in Table 11, the fine-tuned models exhibit improved performance across multiple benchmarks. Notably, the model fine-tuned on HAM10000 achieves a significant overall improvement of 12.9%, indicating that our training method enhances domain-specific cognition without compromising and, in some cases, improves the models' general abilities.

Table 11: General ability evaluation across additional benchmarks. The results demonstrate that our fine-tuned models not only retain their general abilities but also achieve overall improvements compared to the base LLaVA-1.5 model.

| Model | MMStar | SEED-Bench-2 Plus | MMBench | MME (Cognition) | Overall Improvement |
|---|---|---|---|---|---|
| LLaVA-1.5 (Base) | 34.46 | 41.81 | 63.05 | 334.28 | – |
| Trained on CUB-200 | 33.40 | 41.78 | 63.14 | 355.00 | 3.2% ↑ |
| Trained on Stanford Dogs | 34.93 | 40.97 | 63.06 | 365.71 | 8.3% ↑ |
| Trained on FGVC-Aircraft | 35.14 | 40.14 | 63.23 | 348.57 | 2.1% ↑ |
| Trained on PLD | 35.30 | 40.89 | 63.14 | 337.14 | 1.1% ↑ |
| Trained on HAM10000 | 34.46 | 41.11 | 64.08 | 378.21 | 12.9% ↑ |

## F.2 EFFECTIVENESS OF FILTERING STRATEGIES

To evaluate the importance of our reward model-free rejection sampling method described in Section 3.3, we conducted an ablation study comparing our approach with a baseline that does not employ the filtering mechanism. In this baseline, the model generates the most probable responses during each iteration, which are used directly for training without any filtering. Table 12 presents the classification accuracy and cognition score (CS) across four iterations for both the baseline without filtering and our proposed method.

Table 12: Comparison of accuracy and cognition scores (CS) for the baseline without filtering and our proposed method across iterations.

| Dataset | Method | Accuracy (Iter 1) | Accuracy (Iter 2) | Accuracy (Iter 3) | Accuracy (Iter 4) | CS Score |
|---|---|---|---|---|---|---|
| CUB-200 | w/o Filtering | 68.90 | 70.11 | 70.85 | 70.45 | 0.71 |
| | Ours | 80.24 | 83.76 | 84.69 | 85.02 | 0.82 |
| FGVC-Aircraft | w/o Filtering | 76.36 | 76.60 | 77.11 | 76.78 | 0.72 |
| | Ours | 88.78 | 90.91 | 91.42 | 91.99 | 0.79 |
| Stanford Dogs | w/o Filtering | 76.60 | 78.53 | 78.61 | 78.26 | 0.74 |
| | Ours | 85.29 | 86.75 | 86.86 | 86.91 | 0.86 |
| PLD | w/o Filtering | 61.19 | 63.74 | 62.32 | 62.59 | 0.66 |
| | Ours | 75.96 | 92.80 | 96.59 | 97.16 | 0.86 |
| HAM10000 | w/o Filtering | 71.79 | 72.10 | 72.34 | 72.29 | 0.75 |
| | Ours | 79.37 | 82.29 | 83.69 | 85.06 | 0.87 |

From Table 12, it is evident that our filtering strategy significantly enhances both classification accuracy and explanation quality, as measured by the CS score. The baseline without filtering shows marginal improvements initially but fails to achieve comparable performance to our method. This demonstrates the critical role of our filtering mechanism in refining synthetic data and improving the model's performance iteratively.

## F.3 IMPACT OF TEXT ENCODER

Our framework relies on a text embedding model for estimating mutual information during concept selection (Section 3.2). To assess the impact of different text encoders on concept selection accuracy, we compared three models: E5 (Wang et al., 2022), BERT-Large (Devlin, 2018), and BERT-Base (Devlin, 2018). The results are presented in Table 13.

Table 13: Concept selection accuracy using different text encoders.

| Text Encoder | Concept Selection Accuracy (%) |
|---|---|
| E5 | 72.9 |
| BERT-Large | 71.4 |
| BERT-Base | 69.7 |

As shown in Table 13, the E5 model achieves the highest concept selection accuracy. This indicates that using a more powerful text embedding model improves the quality of concept selection, which in turn enhances the effectiveness of our overall framework.

## F.4 CROSS-DATASET TRANSFER PERFORMANCE

We conducted experiments to evaluate the cross-dataset transferability of our method. Specifically, we trained the model on the CUB-200 dataset and evaluated its performance on the Stanford Dogs dataset. The results are shown in Table 14.

Table 14: Cross-dataset transfer performance when training on CUB-200 and evaluating on Stanford Dogs.

| Training Dataset | Evaluation Dataset | Accuracy (%) |
|---|---|---|
| None | Stanford Dogs | 12.20 |
| CUB-200 | Stanford Dogs | 16.60 |
| Stanford Dogs | Stanford Dogs | 86.91 |

The results in Table 14 indicate that training on CUB-200 provides a marginal improvement when evaluated on Stanford Dogs. However, the performance remains significantly lower than when the model is trained directly on Stanford Dogs. This suggests that while our method may improve general cognition to some extent, domain-specific fine-tuning is crucial for achieving high accuracy in specialized tasks.

## F.5 EXPLANATION QUALITY IMPROVEMENT ACROSS ITERATIONS

To assess the benefits of our iterative fine-tuning approach, we evaluated the Cognition Score (CS) of the models across four iterations for various datasets. The results are summarized in Table 15.

Table 15: Cognition Scores (CS) across iterations for various datasets. The CS Improvement represents the percentage increase from Iteration 1 to Iteration 4.

| Dataset | CS Value (Iter 1) | CS Value (Iter 2) | CS Value (Iter 3) | CS Value (Iter 4) | CS Improvement |
|---|---|---|---|---|---|
| CUB-200 | 0.77 | 0.76 | 0.78 | 0.82 | 6.5% ↑ |
| Stanford Dogs | 0.82 | 0.84 | 0.83 | 0.86 | 4.9% ↑ |
| FGVC-Aircraft | 0.78 | 0.78 | 0.78 | 0.79 | 1.3% ↑ |
| PLD | 0.84 | 0.85 | 0.85 | 0.86 | 2.4% ↑ |
| HAM10000 | 0.77 | 0.84 | 0.83 | 0.87 | 13.0% ↑ |
| Chest X-ray | 0.67 | 0.80 | 0.81 | 0.87 | 29.9% ↑ |

As observed in Table 15, the CS scores generally improve over iterations, indicating that our iterative fine-tuning process enhances the explanation quality of the models. The most significant improvements are seen in the HAM10000 and Chest X-ray datasets, with 13.0% and 29.9 % reported.

