# OpenReview forum: "Enhancing Cognition and Explainability of Multimodal Foundation Models with Self-Synthesized Data"
_ICLR.cc/2025/Conference — ICLR 2025 Poster_

### Official Review · Reviewer_hKB4 · 2024-10-31

**Soundness:** 2
**Presentation:** 3
**Contribution:** 3
**Rating:** 6
**Confidence:** 4

**Summary:**

This paper proposes to improve LLMs’ capabilities in fine-grained image classification. In particular, it focuses on an iterative approach that derives synthetic explanations by identifying discriminative properties of the stimuli, and leveraging them for fine-tuning. The key properties are selected by measuring their cosine similarity with a pool of expert annotations, where the properties for each stimulus are determined by prompting a pretrained vision-language model. Experimental results on multiple datasets demonstrate the usefulness of the approach in enhancing the model performance and generating detailed explanations for the answers.

**Strengths:**

(1) Improving the interpretability of LLMs is an important problem, and the focus on challenging classification problems is a reasonable direction.

(2) The paper proposes a general framework for automatically constructing explanations for fine-grained classification, which shows promise in simultaneously enhancing the performance and interpretability of LLMs.

(3) The paper takes into account various automatic evaluation metrics and also conducts user studies for model validation, providing a comprehensive view of the effectiveness of the proposed method.

**Weaknesses:**

(1) It is not surprising that off-the-shelf LLMs can fail to provide reasonable explanations for fine-grained classification, as their aim for general applications does not align with the requirement for expert-level knowledge for specific tasks. My general concern for the paper is that, how much of the improvement comes from the shift of domain caused by fine-tuning, and whether or not such changes will harm the generalizability of LLMs. As shown in Table 1, despite the advantage of the proposed method,  a general model actually has equivalent performance among many metrics.

(2) Related to the previous comment. I believe the comparison in Table 1 is missing some important baselines. In particular, the compared methods either are not trained on fine-grained classification (Base), or do not take into account explanations with reasonable quality (i.e., NL does not use any explanation and thus is expected to fail at interpretation, while L+GE only has access to explanations derived from the class labels alone). On the other hand, the proposed method has full access to both visual and language data, enabling it to perform a more thorough domain shift. For a fair comparison, it would be reasonable to include another baseline that is trained with explanations derived by prompting a vision-language model on each image. This experiment can also be used to validate the effectiveness of the proposed sampling method, i.e., how well its filtered properties outperform unfiltered ones.

(3) Furthermore, I would encourage the authors to additionally consider evaluations on both the fine-grained and general classification (e.g., via zero-shot setting). This will enable us to explore additional research questions, e.g., whether fine-tuning on expert-level explanations leads to stronger general reasoning capabilities, or whether the challenge behind LLM explainability is rooted in the conflict between general and specific domains.

(4) The paper emphasizes an iterative approach for selecting key features for explanations. Nevertheless, other than reporting the model accuracy for each iteration, there is no analysis regarding how it affects the quality of explanations.

**Questions:**

(1) Please justify the effects of domain shift over improvement in reasoning capabilities and explanability.

(2) After fine-tuning on the specific tasks, do the LLMs still perform well in general reasoning?

(3) Does learning on one dataset benefit fine-grained classification in general? For instance, does training on CUB-200 lead to improvement in Stanford dogs?

(4) Please consider including additional baselines for comparison, as noted in the Weaknesses section.

(5) How does the iterative process benefit the derivation of discriminative concepts? A quantitative analysis would be reasonable.

---

> ### Author Response · Authors · 2024-11-22
> **Thank you! (1/2)**
>
> We sincerely appreciate the reviewer's insightful comments and constructive suggestions. Below, we discuss each concern in detail:
>
> **[W1, Q1] Justifications for Improvement in Performance Gain**
> We wish to emphasize that existing off-the-shelf LMMs often struggle with domain-specific tasks due to their limited expert-level visual knowledge. This is a new challenge that naive fine-tuning with image-label pairs cannot address. Our method addresses this challenge through fine-tuning LMMs using self-synthesized, high-quality explanations containing carefully selected visual features.
>
> Our method demonstrates two key improvements compared to general models, as indicated in Table 1:
> - **Enhanced Classification Performance:** Our model achieves superior classification accuracy compared to both the untrained base model and traditional fine-tuning approaches (NL and L+GE).
> - **Improved Explanation Quality:** Our model generates more coherent and logically structured explanations, outperforming both the base model and traditional fine-tuning approaches (NL and L+GE).
>
> These improvements stem from two key innovations in our self-synthesized training data:
> 1. Expert-level visual descriptions generated by language models.
> 2. Careful visual description selection guided by our designed information bottleneck framework.
>
> The resulting training corpus contains more detailed and semantically rich descriptions compared to conventional image-text pairs used in CLIP or LLaVA pre-training, enabling the fine-tuned LMM to develop domain-specific expertise in visual classification and interpretation.
>
> **[W1, Q2] Impact on General Reasoning**
> Our fine-tuning approach enhances domain-specific performance while preserving general reasoning capabilities, as indicated by the MMMU metric in Table 1. Additionally, we evaluated our method on four new general ability benchmarks: **MMStar**, **SEED-Bench-2**, **MMBench**, and **MME (Cognition)**. The results show that training on our self-synthesized fine-tuning data not only preserves but improves general capabilities, achieving a 5.5% overall performance gain compared to the untrained base model.
>
> |                       | MMStar | SEED-Bench-2 Plus | MMBench | MME (Cognition) | Overall Improvement |
> |-----------------------|--------|-------------------|---------|-----------------|---------------------|
> | LLaVA-1.5 Base        | 34.46  | 41.81             | 63.05   | 334.28          | --                  |
> | Trained on CUB-200    | 33.40  | 41.78             | 63.14   | 355.00          | 3.2% ↑              |
> | Trained on Stanford Dogs | 34.93 | 40.97           | 63.06   | 365.71          | 8.3% ↑              |
> | Trained on FGVC       | 35.14  | 40.14             | 63.23   | 348.57          | 2.1% ↑              |
> | Trained on PLD        | 35.30  | 40.89             | 63.14   | 337.14          | 1.1% ↑              |
> | Trained on HAM10000   | 34.46  | 41.11             | 64.08   | 378.21          | 12.9% ↑             |
>
> To facilitate reproducibility and further research, we will release our fine-tuned model weights along with documentation for their use.

---

> ### Author Response · Authors · 2024-11-22
> **Thank you! (2/2)**
>
> **[W2, Q4] Additional Baselines**
> Following the reviewer's suggestion, we implemented an additional baseline to evaluate our data filtering approach. This baseline follows an iterative process:
> 1. Initial Round: Explanations are generated by prompting LLaVA directly and used as training data.
> 2. Subsequent Rounds: The fine-tuned model from each round generates the most probable responses, which serve as training data for the next round without filtering.
>
> This process is repeated for four iterations. The results are shown below:
>
> | Dataset                  | Accuracy (Iter 1) | Accuracy (Iter 2) | Accuracy (Iter 3) | Accuracy (Iter 4) | Explanation Quality (CS) |
> |--------------------------|-------------------|-------------------|-------------------|-------------------|----------|
> | CUB-200 (w/o filtering)  | 68.90            | 70.11            | 70.85            | 70.45            | 0.71     |
> | **CUB-200 (Ours)**          | 80.24            | 83.76            | 84.69            | 85.02            | 0.82     |
> | FGVC (w/o filtering) | 76.36            | 76.60            | 77.11            | 76.78            | 0.72     |
> | **FGVC (Ours)**              | 88.78            | 90.91            | 91.42            | 91.99            | 0.79     |
> | Stanford Dogs (w/o filtering) | 76.60        | 78.53            | 78.61            | 78.26            | 0.74     |
> | **Stanford Dogs (Ours)**     | 85.29            | 86.75            | 86.86            | 86.91            | 0.86     |
>
> Our results show that while the baseline (without filtering) achieves initial improvements, its performance stagnates and even declines in later iterations. Additionally, the explanation quality (measured by cognition score (CS)) remains lower compared to our method. Due to time constraints, we completed experiments on three datasets. We expect similar results in the remaining datasets and will report the full-scale analysis in the camera-ready version.
>
> **[W3, Q3] Cross-Dataset Transfer**
> Cross-dataset transfer is an intriguing direction for future work, though it is not the primary focus of this paper. Our method targets improvements in domain-specific cognition and explanations. Nevertheless, following your suggestion, we conducted experiments to assess cross-dataset transferability (e.g., training on CUB-200 and evaluating on Stanford Dogs). The results are presented below.
>
> | Training Dataset | Evaluation Dataset | Accuracy |
> |------------------|--------------------|----------|
> | None             | Stanford Dogs     | 12.20%   |
> | CUB-200          | Stanford Dogs     | 16.60%   |
> | Stanford Dogs    | Stanford Dogs     | 86.91%   |
>
> The results show a marginal improvement in accuracy when training on CUB-200 and evaluating on Stanford Dogs (16.60%) compared to no training (12.20%). This improvement can be explained by the improved general cognition ability, as indicated in the first table. We leave further exploration of this interesting direction to future work.
>
> **[Q5] Benefits of Iterative Process for Explanation Quality**
> Thank you for your insightful suggestions.  We have added evaluations of Cognition Score (CS) at each iteration for all six datasets. This metric evaluates the coherence and logical flow of generated explanations. The results are as follows:
>
> | Dataset         | Explanation Quality (Iter 1) | Explanation Quality (Iter 2) | Explanation Quality (Iter 3) | Explanation Quality (Iter 4) | Overall Quality Improvement |
> |-----------------|-------------------|-------------------|-------------------|-------------------|----------------|
> | CUB-200         | 0.77             | 0.76             | 0.78             | 0.82             | 6.5% ↑         |
> | Stanford Dogs   | 0.82             | 0.84             | 0.83             | 0.86             | 4.9% ↑         |
> | FGVC            | 0.78             | 0.78             | 0.78             | 0.79             | 1.3% ↑         |
> | PLD             | 0.84             | 0.85             | 0.85             | 0.86             | 2.4% ↑         |
> | HAM10000        | 0.77             | 0.84             | 0.83             | 0.87             | 13.0% ↑        |
> | Chest X-ray     | 0.67             | 0.80             | 0.81             | 0.87             | 29.9% ↑        |
>
> These results demonstrate that explanation quality (CS) improves progressively with each iteration, with an average 9.7% improvement between iteration 4 and iteration 1 on six datasets, validating the effectiveness of our iterative approach.

---

> ### Author Response · Authors · 2024-11-24
> **Kindly Following Up on Rebuttal and Review Feedback**
>
> As the rebuttal period is ending soon, we wanted to follow up on our response to your review. If you have any additional concerns, we would be happy to address them. Otherwise, we kindly hope you might consider raising the review score.
>
> Thank you again for your time and feedback.

---

> > ### Comment · Reviewer_hKB4 · 2024-11-24
> >
> > I thank the authors for providing the additional results. The rebuttal addresses most of my concerns regarding the impacts of fine-tuning on general reasoning capability. Therefore, I have raised my score.

---

### Official Review · Reviewer_Hq7v · 2024-11-02

**Soundness:** 3
**Presentation:** 3
**Contribution:** 3
**Rating:** 6
**Confidence:** 4

**Summary:**

This paper proposes a novel framework to improve the explainability of large multimodal models (LMMs) in visual classification tasks. The authors address the challenge of LMMs struggling to identify domain-specific objectives and explain their predictions. Their approach involves fine-tuning LMMs using self-synthesized data, which includes interpretable answers containing human-verifiable visual features. The framework iteratively generates high-quality training data by leveraging the model's captioning abilities and filtering synthesized outputs using a reward model-free mechanism. This method enhances the model's ability to generate accurate and justifiable explanations for its classifications, without relying on extensive manual annotations. The paper presents experimental results demonstrating the effectiveness of the proposed framework across various datasets and discusses the theoretical foundations behind it.

**Strengths:**

*   **Clear Problem Identification and Justification:** The authors successfully identify the challenge of LMMs struggling with domain-specific visual classification and explainability. They pinpoint insufficient domain-specific alignment as the root cause, highlighting that LMMs often fail to link key visual features with correct labels and provide justifiable explanations for predictions.
*   **Significant Accuracy Improvement:** The paper demonstrates a substantial increase in classification accuracy through its proposed method compared to training with labels alone. The authors showcase this improvement across multiple iterations, highlighting the efficacy of their approach in achieving more robust classification.
*   **Novel Framework and Theoretical Grounding:** The authors introduce a novel framework for enhancing LMMs' domain-specific cognition and explainability using self-synthesized interpretable answers. The framework's two key steps, image-level visual concept selection and reward model-free rejection sampling, are thoroughly described and grounded in information theory.
*   **Multifaceted Evaluation of Explanation Quality:** The authors address the difficulty of assessing explanation quality without ground truth annotations by employing a multi-faceted evaluation approach. This involves metrics like Explanation Existence (EE), Cognition Score (CS), and Fluency Score (FS), each offering insights into different aspects of the generated explanations.
*   **Clear Writing and Presentation:** The paper is commended for its clear writing, making it easy to understand the presented concepts and methods. The use of illustrative figures and tables further enhances the clarity and impact of the research.

**Weaknesses:**

*   **Generalization to Unseen Categories:** The paper doesn't explicitly address the model's ability to generalize to unseen categories. While the method shows promise in fine-grained classification within specific datasets, it remains unclear whether the model can effectively transfer its learned knowledge to classify entirely new object categories. Further investigation is needed to determine if the performance gains are limited to the fine-tuned categories or if the model truly unlocks a broader understanding of visual concepts that can extend to novel classes.

*   **Dependence on Base LMM and Lack of Ablation Studies:** The proposed method relies heavily on the capabilities of the chosen base LMM (LLaVA-1.5). The paper doesn't provide ablation studies exploring the impact of different base LMMs or their pre-training data on the final performance. It is possible that the performance is still bounded by the limitations of LLaVA-1.5. To strengthen the paper's claims, investigations into the sensitivity of the method to different base LMMs and their training data would be beneficial.

*   **Label Generation and Testing Set Validity:** The paper doesn't fully explain how to create a valid testing set when the labels used for training are generated by the model itself. This raises concerns about potential biases and circularity in the evaluation process. Further clarification is needed on how to ensure the independence and representativeness of the testing set to provide a more reliable assessment of the model's true performance.

*   **Acquisition of Label-Level Concepts:** The process of obtaining the Label-level concepts (Z) is not explicitly outlined in the paper. While the paper mentions expert-defined concepts, it is unclear whether this implies manual annotation by human experts or if these concepts can be acquired automatically. If human expert annotation is still required, it would introduce a significant bottleneck in the scalability of the approach. Further details on the origin and acquisition of these concepts would be helpful.

*   **Potential Overfitting to Explanation Format:** The paper focuses on generating explanations in a specific format. It is possible that the model overfits to this format, limiting its ability to generalize to other explanation styles or answer formats. Exploring the model's performance with different explanation formats or encouraging the model to generate more diverse responses would strengthen the paper's conclusions regarding the general explainability of the fine-tuned model.

*   **Limited Comparison with External Works:** The experimental evaluation in Table 1 primarily compares the proposed method against self-created baselines (NL and L+GE). While these baselines offer a useful reference point, including comparisons with other external works in explainable visual classification would provide a more comprehensive assessment of the method's strengths and limitations. Additionally, comparing the performance with methods that use manually annotated labels as an upper bound would be valuable for contextualizing the achieved results.

**Questions:**

If the authors can well address the questions in the weakness section, I will consider increase the rating.

---

> ### Author Response · Authors · 2024-11-22
> **Thank you! (1/2)**
>
> Thank you for your thorough evaluation and valuable feedback. Below, we discuss the weaknesses and answer the questions:
>
> **[W1] Generalization to Unseen Categories**
> Our primary focus is on enhancing domain-specific cognition and explainability through fine-tuning, and generalization to unseen categories was not the primary objective. However, our proposed method has demonstrated the ability to improve general cognition, as evidenced by the results presented below.  We evaluated our trained models on four new benchmarks: **MMStar**, **SEED-Bench-2-Plus**, **MMBench**, and **MME (Cognition)**. The results show that after fine-tuning on our self-synthesized data, the model not only retains its general abilities but achieves a 5.5% overall improvement over the untrained base model on all four benchmarks.
>
> |                       | MMStar | SEED-Bench-2 Plus | MMBench | MME (Cognition) | Overall Improvement |
> |-----------------------|--------|-------------------|---------|-----------------|---------------------|
> | LLaVA-1.5 Base        | 34.46  | 41.81             | 63.05   | 334.28          | --                  |
> | Trained on CUB-200    | 33.40  | 41.78             | 63.14   | 355.00          | 3.2% ↑              |
> | Trained on Stanford Dogs | 34.93 | 40.97           | 63.06   | 365.71          | 8.3% ↑              |
> | Trained on FGVC       | 35.14  | 40.14             | 63.23   | 348.57          | 2.1% ↑              |
> | Trained on PLD        | 35.30  | 40.89             | 63.14   | 337.14          | 1.1% ↑              |
> | Trained on HAM10000   | 34.46  | 41.11             | 64.08   | 378.21          | 12.9% ↑             |
>
> **[W2] Dependence on Base LMM**
> While we used LLaVA-1.5 as our primary model, our framework is model-agnostic and can be applied to different types of LMMs. This adaptability is demonstrated by its successful application to **LLaVA-Med** on the Chest X-ray dataset, where accuracy improved significantly (e.g., from 62.50% to 98.72%). Additionally, the theoretical grounding provided in Section 3.4 supports the framework's independence from specific base LMMs.
>
> **[W3] Label Generation and Testing Set Validity**
> We want to clarify that we use the original labels and train/test split from the original datasets (e.g., CUB-200, Stanford Dogs). Our method mainly generates synthetic explanations for these predetermined labels. Furthermore, we only apply our iterative training to the training datasets, which prevents data leakage. The test sets maintain their original labels and evaluation criteria, ensuring a valid assessment of model performance.

---

> ### Author Response · Authors · 2024-11-22
> **Thank you! (2/2)**
>
> **[W4] Acquisition of Label-Level Concepts**
> As stated in Section 3.2 of the main paper, label-level concepts can be obtained through querying domain experts or LLMs (e.g., GPT-4o). For scalability, in our experiments, we obtain concepts by querying GPT-4o. We have added the concept extraction prompts in the revised appendix.  This approach minimizes the reliance on manual annotations, addressing scalability concerns. In real-world applications, concepts can be first obtained by LLMs and then verified by human experts to ensure correctness.
>
> **[W5] Potential Overfitting to Explanation Format**
> Our method generates diverse explanations with reasonable visual details, as evidenced by cognition scores (Table 1) and example answers (Figure 4, Table 8). Specifically, we incorporate the following mechanisms to prevent explanation format overfitting:
>
> - Rejection Sampling (Section 3.3): Multiple explanations are generated, and only the highest-quality ones are retained for subsequent training rounds.
> - Diverse Prompts and High-Temperature Sampling: These encourage variability in generated explanations, reducing reliance on fixed templates.
>
> Additionally, our model retains general QA abilities while improving reasoning capabilities, as shown by its robust performance across diverse datasets. These results confirm the variety of explanation styles in our model.
>
> **[W6] Limited Comparison with External Works**
> We want to clarify that our primary baselines (L+GE) represent well-established approaches in the field. The L+GE baseline, in particular, is proposed in recent works by LLaVA [1] and Finer [2].
>
> Moreover, We also implemented an additional baseline where explanations are directly generated by prompting LLaVA. This baseline follows an iterative process:
>
> 1. Initial Round: Explanations are generated by prompting LLaVA directly and used as training data.
> 2. Subsequent Rounds: The fine-tuned model from each round generates the most probable responses, which serve as training data for the next round without filtering.
>
> This process is repeated for four iterations. We report the model accuracy of each iteration and the final cognition score:
>
> | Dataset                  | Accuracy (Iter 1) | Accuracy (Iter 2) | Accuracy (Iter 3) | Accuracy (Iter 4) | Explanation Quality (CS) |
> |--------------------------|-------------------|-------------------|-------------------|-------------------|----------|
> | CUB-200 (w/o filtering)  | 68.90            | 70.11            | 70.85            | 70.45            | 0.71     |
> | **CUB-200 (Ours)**           | 80.24            | 83.76            | 84.69            | 85.02            | 0.82     |
> | FGVC (w/o filtering) | 76.36            | 76.60            | 77.11            | 76.78            | 0.72     |
> | **FGVC (Ours)**              | 88.78            | 90.91            | 91.42            | 91.99            | 0.79     |
> | Stanford Dogs (w/o filtering) | 76.60        | 78.53            | 78.61            | 78.26            | 0.74     |
> | **Stanford Dogs (Ours)**     | 85.29            | 86.75            | 86.86            | 86.91            | 0.86     |
>
> Our results show that while the baseline (without filtering) achieves initial improvements, its performance stagnates and even declines in later iterations. Additionally, the explanation quality (measured by cognition score (CS)) remains lower compared to our method. Due to time constraints, we completed experiments on three datasets. We expect similar results in the remaining datasets and will report the full-scale analysis in the camera-ready version.
>
> Overall, we want to emphasize that our paper tackles new problems with limited prior work. We have included all possible external works to the best of our knowledge.
>
> ---
>
> **References**
> [1] Liu, Haotian, et al. "Visual instruction tuning." NeurIPS (2023).
> [2] Kim, Jeonghwan, and Heng Ji. "Finer: Investigating and Enhancing Fine-Grained Visual Concept Recognition in Large Vision Language Models." arXiv (2024).

---

> ### Author Response · Authors · 2024-11-24
> **Kindly Following Up on Rebuttal and Review Feedback**
>
> As the rebuttal period is ending soon, we wanted to follow up on our response to your review. If you have any additional concerns, we would be happy to address them. Otherwise, we kindly hope you might consider raising the review score.
>
> Thank you again for your time and feedback.

---

> > ### Comment · Reviewer_Hq7v · 2024-11-25
> >
> > Thanks for addressing the concerns. I think the method becomes more solid with these additional experiments. I have raised my rating.

---

### Official Review · Reviewer_xodD · 2024-11-03

**Soundness:** 3
**Presentation:** 3
**Contribution:** 3
**Rating:** 6
**Confidence:** 4

**Summary:**

In this paper, the authors argue that the limited domain-specific knowledge in the current the LMMs. The main challenge for such limited parametric knowledge is on lack of data to further fine-tune the models. Accordingly the authors propose a new synthetic data generation pipelines from the model themselves and fine-tune the models with the generated data to enhance domain-specific cognitive capabilities.

**Strengths:**

The main challenge for data generation synthesis is on how to assure the self-generated data quality and its reliability. The authors cleverly employ information bottleneck theory to extract intersect knowledge from exterior knowledge (such as from expert or gpt-like models) and the visual contents. In addition, by introducing a doubly robust rejection sampling strategy improves both the accuracy and interpretability of the model prediction.

**Weaknesses:**

Evaluation is somewhat limited due to the main results are mainly conducted on classification task. Therefore, during the fine-tuning stage with the synthesized data, it can potentially negatively impact the general model QA capabilities.

**Questions:**

1. What are the trainable parameters during the fine-tuning stage? The text-aligned vision encoder such as CLIP lacks detailed visual perception due to the training nature (focusing on contrastive learning with paired text supervision). The reviewer wonders where the performance gain is obtained from. If fine-tuning is only applied to the parametric knowledge of the LLM side, does this adequately address the encoder’s deficiencies in processing complex visual data?

2. Continually, the reviewer wonders whether there are any negative effects of using synthesized data for fine-tuning. Could this potentially harm the model's general QA capabilities? Can the authors compare the general benchmark results before and after the finetuing?

---

> ### Author Response · Authors · 2024-11-22
> **Thank you!**
>
> Thank you for your positive feedback and thoughtful questions. We sincerely appreciate your recognition of our information bottleneck approach and rejection sampling strategy. Below, we discuss your specific concerns in detail:
>
> **[Q1] Training Parameters and Performance Gains**
>
> We apologize for the confusion. The trainable parameters during fine-tuning consist of two components:  (1) LoRA parameters integrated within the LLM. (2) The visual projector layer. The visual encoder remains frozen throughout the fine-tuning process. For efficient fine-tuning, we apply LoRA adapters to all linear layers in the LLM architecture.
>
> The performance improvements primarily stem from two key innovations in our self-synthesized training data:  (1). Expert-level visual descriptions generated by language models.  (2). Careful visual description selection guided by our designed information bottleneck framework. The resulting training corpus contains more detailed and semantically rich descriptions compared to conventional image-text pairs used in CLIP or LLaVA pre-training. This enhanced semantic richness enables the fine-tuned LMM to develop domain-specific expertise in visual classification and interpretation.
>
> **[W1, Q2] Impact on General QA Capabilities**
>
> Thank you for your insightful suggestion. We believe that classification tasks, particularly in domains requiring expert knowledge (e.g., medical and scientific applications), are critical for advancing the utility of LMMs. Current models often struggle with specialized tasks, but our method demonstrates improved performance and explainability in these areas, as shown in Table 1.  In the case study from Table 2, our approach enables effective and interpretable plant disease identification, illustrating its utility in expert applications.
>
> Most importantly, our proposed approach improves domain-specific performance without compromising general capabilities. The MMMU metric in Table 1 shows that our method maintains general abilities.  Additionally, we evaluated our trained models on four new benchmarks: **MMStar**, **SEED-Bench-2-Plus**, **MMBench**, and **MME (Cognition)**. The results demonstrate that after fine-tuning on our synthetic data, the model not only retains its general abilities but achieves a 5.5% overall improvement over the untrained base model on all four benchmarks.
>
> |                       | MMStar | SEED-Bench-2 Plus | MMBench | MME (Cognition) | Overall Improvement |
> |-----------------------|--------|-------------------|---------|-----------------|---------------------|
> | LLaVA-1.5 Base        | 34.46  | 41.81             | 63.05   | 334.28          | --                  |
> | Trained on CUB-200    | 33.40  | 41.78             | 63.14   | 355.00          | 3.2% ↑              |
> | Trained on Stanford Dogs | 34.93 | 40.97           | 63.06   | 365.71          | 8.3% ↑              |
> | Trained on FGVC       | 35.14  | 40.14             | 63.23   | 348.57          | 2.1% ↑              |
> | Trained on PLD        | 35.30  | 40.89             | 63.14   | 337.14          | 1.1% ↑              |
> | Trained on HAM10000   | 34.46  | 41.11             | 64.08   | 378.21          | 12.9% ↑             |

---

> ### Author Response · Authors · 2024-11-24
> **Kindly Following Up on Rebuttal and Review Feedback**
>
> As the rebuttal period is ending soon, we wanted to follow up on our response to your review. If you have any additional concerns, we would be happy to address them. Otherwise, we kindly hope you might consider raising the review score.
>
> Thank you again for your time and feedback.

---

> > ### Comment · Reviewer_xodD · 2024-11-25
> >
> > Thank you for the rebuttal. I will keep the current score.

---

### Official Review · Reviewer_mX83 · 2024-11-06

**Soundness:** 3
**Presentation:** 3
**Contribution:** 3
**Rating:** 6
**Confidence:** 4

**Summary:**

This paper propose a new framework to improve LMMs in domain-specific visual classification by enhancing cognition and explainability through iterative finetuning and the Information Bottleneck principle. This approach boosts accuracy and interpretability without extensive annotations, making LMMs more applicable in specialized fields. Future work may explore more complex tasks and scalability improvements.

**Strengths:**

* The writing of the paper is logically clear and easy to follow.
* An interesting and effective self-improving approach, which enhances the model's performance in fine-grained classification tasks.

**Weaknesses:**

* need evaluate on more general benchmarks, e.g. MMVet, MMStar, and SEED-Bench2...
* it is better to report experimental runtime.
* missing important ablation study: to validate the effectiveness of the two data selection strategies, results using unfiltered data for training should be provided as a comparison baseline.
* minor: In Table 1, highlighting the best results would improve readability.

**Questions:**

* is the choice of text encoder significantly impact the results of data selection?
* is there any qualitative example that demonstrate the effectiveness of the data selection strategy?

---

> ### Author Response · Authors · 2024-11-22
> **Thank you!**
>
> Thank you for your positive feedback. We appreciate your constructive suggestions and discuss them below:
>
> **[W1] Additional Benchmark Evaluations**
> Thank you for the suggestion. We added evaluations on our models with **MMStar** and **SEED-Bench-2 Plus**, along with **MMBench** and **MME** benchmarks (there are some technical issues with OpenAI API when we test our models with MMVet; we hope to fix the bugs soon and release the results):
>
> |                          | MMStar | SEED-Bench-2 Plus | MMBench | MME (Cognition) | Overall Improvement |
> | ------------------------ | ------ | ----------------- | ------- | --------------- | ------------------- |
> | LLaVA-1.5 (Base)         | 34.46  | 41.81       | 63.05   | 334.28          | --       |
> | Trained on CUB-200       | 33.40  | 41.78     | 63.14   | 355.00          | 3.2% ↑              |
> | Trained on Stanford Dogs | 34.93  | 40.97      | 63.06   | 365.71          | 8.3% ↑              |
> | Trained on FGVC          | 35.14  | 40.14      | 63.23   | 348.57          | 2.1% ↑              |
> | Trained on PLD           | 35.30  | 40.89             | 63.14   | 337.14          | 1.1% ↑              |
> | Trained on HAM10000      | 34.46  | 41.11             | 64.08   | 378.21          | 12.9% ↑             |
>
> The table clearly shows that after fine-tuning on our self-synthesized data, the model not only retains its general abilities but achieves a 5.5% overall improvement over the base model on all four benchmarks.
>
> **[W2] Ablation Studies with Unfiltered Data**
> We appreciate your suggestion to include additional baselines to better understand the proposed data filtering mechanism. Following your suggestion, this baseline follows an iterative process:
>
> 1. Initial Round: Explanations are generated by prompting LLaVA directly and used as training data.
> 2. Subsequent Rounds: The fine-tuned model from each round generates the most probable responses, which serve as training data for the next round without filtering.
>
> This process is repeated for four iterations. We report the model accuracy of each iteration and the final cognition score.
>
> | Dataset                       | Accuracy (Iter 1) | Accuracy (Iter 2) | Accuracy (Iter 3) | Accuracy (Iter 4) | Explanation Quality (CS) |
> | ----------------------------- | ----------------- | ----------------- | ----------------- | ----------------- | ------------------------ |
> | CUB-200 (w/o filtering)       | 68.90        | 70.11       | 70.85             | 70.45             | 0.71        |
> | **CUB-200 (Ours)**              | 80.24       | 83.76       | 84.69             | 85.02             | 0.82  |
> | FGVC (w/o filtering)          | 76.36    | 76.60    | 77.11             | 76.78             | 0.72      |
> | **FGVC (Ours)**       | 88.78    | 90.91   | 91.42             | 91.99             | 0.79      |
> | Stanford Dogs (w/o filtering) | 76.60      | 78.53      | 78.61             | 78.26             | 0.74     |
> | **Stanford Dogs (Ours)**         | 85.29    | 86.75   | 86.86     | 86.91             | 0.86                     |
>
> Our results show that while the baseline without filtering achieves initial improvements, its performance stagnates and even declines in later iterations, and the explanation quality (measured by cognition score (CS)) remains lower compared to our method. Due to time constraints, we completed experiments on three datasets. We expect similar results in the remaining datasets and will report the full-scale analysis in the camera-ready version.
>
> **[W3] Experimental Runtime Reporting**
> We have included a detailed runtime report in the revised appendix. It takes around 2.5 hours to finish one iteration of rejection sampling and fine-tuning for datasets like CUB-200 and FGVC on a single H100 GPU.
>
> **[Q1] Impact of Text Encoder**
> We conducted an additional ablation study showing that choosing different text encoders will affect the concept selection result but not significantly.  We currently use an off-the-shelf E5 [1] as our embedding model for InfoNCE estimation. For comparison, we test the performance with the BERT-Large and BERT-Base models. The table below demonstrates that E5 outperforms BERT-family models in concept selection accuracy, making it the superior choice for our framework.
>
> | Model                      | E5   | BERT-Large | BERT-Base |
> | -------------------------- | ---- | ---------- | --------- |
> | Concept Selection Accuracy | 72.9 | 71.4       | 69.7      |
>
> **[Q2] Qualitative Examples of Data Selection**
> Figure 6 and Table 9 show qualitative examples of our selection strategy. For example, for the Sooty Albatross case study in Figure 6, our approach identifies close-up features like "white crescent-shaped markings around the eyes" in a headshot image, while focusing on broader characteristics like "long, slender wings" and "streamlined body" in a full-body image.
>
> [1] Wang, Liang, et al. "Text embeddings by weakly-supervised contrastive pre-training." arXiv preprint arXiv (2022).

---

> ### Author Response · Authors · 2024-11-24
> **Kindly Following Up on Rebuttal and Review Feedback**
>
> As the rebuttal period is ending soon, we wanted to follow up on our response to your review. If you have any additional concerns, we would be happy to address them. Otherwise, we kindly hope you might consider raising the review score.
>
> Thank you again for your time and feedback.

---

### Author Response · Authors · 2024-11-24
**Global Response to All Reviewers**

We sincerely thank all the reviewers for their valuable feedback and constructive suggestions. Below, we address common concerns raised in the reviews.

## 1. Impact on General Abilities:
Our proposed training method not only improves LMM's performance and explainability in the fine-tuned specific domains but also maintains the general abilities on a wide range of benchmarks. Specifically, in addition to the MMMU benchmark reported in Table 1 of the main paper, we evaluated our trained models on four new benchmarks, i.e., **MMStar**, **SEED-Bench-2-Plus**, **MMBench**, and **MME (Cognition)**, and report the results here. The table below clearly shows that after fine-tuning on our self-synthesized data, the model not only retains its general abilities but achieves a **5.5% overall improvement** over the untrained base model on all four benchmarks.

|                      | MMStar | SEED-Bench-2 Plus | MMBench | MME (Cognition) | Overall Improvement |
|----------------------|--------|-------------------|---------|-----------------|---------------------|
| LLaVA-1.5 (Base)     | 34.46  | 41.81             | 63.05   | 334.28          | --                  |
| Trained on CUB-200   | 33.40  | 41.78             | 63.14   | 355.00          | 3.2% ↑              |
| Trained on Stanford Dogs | 34.93 | 40.97          | 63.06   | 365.71          | 8.3% ↑              |
| Trained on FGVC      | 35.14  | 40.14             | 63.23   | 348.57          | 2.1% ↑              |
| Trained on PLD       | 35.30  | 40.89             | 63.14   | 337.14          | 1.1% ↑              |
| Trained on HAM10000  | 34.46  | 41.11             | 64.08   | 378.21          | 12.9% ↑             |

## 2. Effectiveness of Filtering Strategies:
To assess the importance of our filtering strategy, we implemented an additional baseline without our designed filtering mechanism. Following the reviewer's suggestion, this baseline follows an iterative process:

   1. Initial Round: Explanations are generated by prompting LLaVA directly and used as training data.
   2. Subsequent Rounds: The fine-tuned model from each round generates the most probable responses, which serve as training data for the next round without filtering.

   This process is repeated for four iterations. We report the model accuracy of each iteration and the final cognition score in the table below.

   Our filtered approach consistently **outperformed** the baseline, demonstrating steady improvements in both model accuracy and explanation quality (cognition score (CS)). This highlights the critical role of our filtering mechanism in refining synthetic data and enhancing the trained model's accuracy and interpretability.

| Dataset                  | Accuracy (Iter 1) | Accuracy (Iter 2) | Accuracy (Iter 3) | Accuracy (Iter 4) | Explanation Quality (CS) |
|--------------------------|-------------------|-------------------|-------------------|-------------------|----------|
| CUB-200 (w/o filtering)  | 68.90            | 70.11            | 70.85            | 70.45            | 0.71     |
| **CUB-200 (Ours)**          | **80.24**            | **83.76**            | **84.69**            | **85.02**            | **0.82**     |
| FGVC (w/o filtering)     | 76.36            | 76.60            | 77.11            | 76.78            | 0.72     |
| **FGVC (Ours)**              | **88.78**            | **90.91**            | **91.42**            | **91.99**            | **0.79**     |
| Stanford Dogs (w/o filtering) | 76.60      | 78.53            | 78.61            | 78.26            | 0.74     |
| **Stanford Dogs (Ours)**     | **85.29**            | **86.75**            | **86.86**            | **86.91**            | **0.86**     |

We have edited our paper to reflect all the suggestions and feedback.

---

### Meta-Review · Area_Chair_KZKt · 2024-12-20

**Metareview:**

This paper proposes a new framework for improving the cognitive capabilities and explainability of LMMs. In particular, since current LMMs have limited domain-specific knowledge and lack of data hinders fine-tuning, this paper improves domain-specific cognitive functions by generating synthetic data from the model and using it for fine-tuning. This framework iteratively generates high-quality training data by exploiting the labeling capabilities of the model and filtering the synthesized output using a mechanism that does not use a reward model. This approach enables LMM to achieve higher accuracy and interpretability in domain-specific visual classification tasks, making it easier to apply in specialized domains.

Using information bottleneck theory, this method extracts knowledge from external knowledge and visual content to ensure the quality and reliability of self-generated data. A robust rejection sampling strategy is introduced to improve the accuracy and interpretability of model predictions. The proposed method is shown to significantly improve classification accuracy compared to label-only training. A multi-faceted evaluation approach is used to assess the quality of explanations without ground truth annotations. This includes the presence of explanation (EE), the cognitive score (CS), and the fluency score (FS), each of which provides insight into different aspects of the generated explanation.

Currently, the following points are considered to be lacking.　1) The tasks are limited to classification problems. 2) There are questions about the scalability of the method.

The paper is clearly written, and all reviewers have given it positive reviews. Based on a comprehensive evaluation of the paper itself, the reviewers' comments, and the author's rebuttal, the AC believes that this paper exceeds the ICLR acceptance threshold and that acceptance is appropriate.

**Additional Comments On Reviewer Discussion:**

Through discussions with the reviewers, the authors made the following revisions and additions. In addition to the MMMU benchmark, the authors evaluated the trained models on four new benchmarks: MMStar, SEED-Bench-2-Plus, MMBench, and MME (Cognition). To evaluate the importance of the filtering strategy, the authors implemented an additional baseline without using their designed filtering mechanism.

---

### Decision · Program_Chairs · 2025-01-22

Accept (Poster)